# Novel immunomodulatory properties of adenosine analogs promote their antiviral activity against SARS-CoV-2

Giulia Monticone [1✉], Zhi Huang [1], Peter Hewins [2], Thomasina Cook [2], Oygul Mirzalieva[3], Brionna King[1], Kristina Larter[1], Taylor Miller-Ensminger[2], Maria D Sanchez-Pino[1,4], Timothy P Foster[5], Olga V Nichols[5], Alistair J Ramsay [5], Samarpan Majumder [1], Dorota Wyczechowska [4], Darlene Tauzier[6,7], Elizabeth Gravois[6,7], Judy S Crabtree [1,6], Jone Garai[4], Li Li[4], Jovanny Zabaleta [4], Mallory T Barbier[4], Luis Del Valle [4,7], Kellie A Jurado[2] & Lucio Miele [1]

## Abstract

**The COVID-19 pandemic reminded us of the urgent need for new antivirals to control emerging infectious diseases and potential future pandemics. Immunotherapy has revolutionized oncology and could complement the use of antivirals, but its application to infectious diseases remains largely unexplored. Nucleoside analogs are a class of agents widely used as antiviral and anti-neoplastic drugs. Their antiviral activity is generally based on interference with viral nucleic acid replication or transcription. Based on our previous work and computer modeling, we hypothesize that antiviral adenosine analogs, like remdesivir, have previously unrecognized immunomodulatory properties which contribute to their therapeutic activity. In the case of remdesivir, we here show that these properties are due to its metabolite, GS-441524, acting as an Adenosine A2A Receptor antagonist. Our findings support a new rationale for the design of next-generation antiviral agents with dual - immunomodulatory and intrinsic - antiviral properties. These compounds could represent game-changing therapies to control emerging viral diseases and future pandemics.**

**Keywords** Antivirals; Adenosine Analogs; Remdesivir; Adenosine; Immunotherapy
**Subject Categories** Immunology; Microbiology, Virology & Host Pathogen Interaction; Pharmacology & Drug Discovery

## Introduction

The COVID-19 pandemic has caused over 6 million deaths worldwide (WorldHealthOrganization, 2020) what will be remembered as an unprecedented global emergency. More than 3 years after the emergence of SARS-CoV-2, remdesivir (REM) and Paxlovid are the only FDA-approved antiviral drugs for COVID-19 (Beigel et al, 2020; Liu et al, 2023). Vaccines contributed to control the pandemic; however, their use has encountered several obstacles, including low immunogenicity of SARS-CoV-2, the emergence of new variants and vaccine hesitancy (Mohamed et al, 2022). Given the critical challenges that society has faced to control COVID-19, it is imperative to design effective next-generation antivirals for coronaviruses and other future emerging infectious diseases.

Current efforts to design antivirals for infectious diseases have been focusing on approaches that directly interfere with pathogens, including blocking viral replication or cell entry (Clercq and Li, 2016). However, given the sometime drastic impairment of the immune system in many viral diseases, like COVID-19 (Li et al, 2022; Shi et al, 2020), antiviral therapies alone do not always guarantee a full recovery. Chronic viral persistence and reactivation of endogenous dormant viruses such as EBV are among the theories advanced to explain "post-acute sequelae of SARS-CoV-2 infection" (PASC or "Long COVID"). Transplant patients, who must take immunosuppressive drugs to prevent rejection, have a higher risk of severe outcomes in acute COVID-19 (Massie et al, 2022; Udomkarnjananun et al, 2021). Several studies reported lymphopenia, CD8 + T-cell impairment and excessive neutrophil activation in COVID-19 cases (Barnes et al, 2020; Diao et al, 2020; He et al, 2020; Huang et al, 2020; Liao et al, 2020; Zheng et al, 2020). Accordingly, patients with COVID-19 were found to have an

[1]Department of Genetics, School of Medicine, Louisiana State University Health Sciences Center, New Orleans, LA, USA. [2]Department of Microbiology, Perelman School of Medicine, University of Pennsylvania, Philadelphia, PA, USA. [3]Department of Biochemistry and Molecular Biology, School of Medicine, Louisiana State University Health Sciences Center, New Orleans, LA, USA. [4]Department of Interdisciplinary Oncology, School of Medicine, Louisiana State University Health Sciences Center, New Orleans, LA, USA. [5]Department of Microbiology, Immunology & Parasitology, School of Medicine, Louisiana State University Health Sciences Center, New Orleans, LA, USA. [6]Precision Medicine Program, Louisiana State University Health Sciences Center, New Orleans, LA, USA. [7]Department of Pathology, School of Medicine, Louisiana State University Health Sciences Center, New Orleans, LA, USA. ✉E-mail: gmonti@lsuhsc.edu

elevated neutrophil-to-lymphocyte ratio (NLR) and this correlated with disease severity (Liu et al, 2020a; Liu et al, 2020b). Importantly, rebounds in T-cell numbers are observed in patients with improved conditions (Diao et al, 2020; He et al, 2020; Liao et al, 2020; Zheng et al, 2020), suggesting that the prognosis of COVID-19, and likely of other viral diseases, directly correlates with the ability of the host immune system to respond to the pathogen.

In the past decade, immunotherapy has revolutionized the field of oncology, shifting the focus from therapies that directly interfere with tumor cells to therapies that promote anti-tumor immune responses. Immunotherapy for infectious diseases remains a largely unexplored field. Adenosine is a physiological immunosuppressive metabolite that controls immune responses by activating GPCR adenosine receptors on immune cells (Antonioli et al, 2013; Ohta 2016). In tumors and some infectious diseases, adenosine is overproduced (Antonioli et al, 2013; da Silva et al, 2022; Drygiannakis et al, 2011; Ohta 2016; Passos et al, 2018), thus inappropriately suppressing protective immune responses. Our group and others have shown that blocking the adenosine-mediated activation of the Adenosine A2A Receptor (A2AR) with A2AR antagonists boosts CD8 + T-cell functions and protects CD8 + T-cells from exhaustion and immune suppression (Leone et al, 2018; Monticone et al, 2022; Sorrentino et al, 2019; Willingham et al, 2018). Other studies suggest that A2AR blockade might also positively modulate other immune cells, including CD4+ effector T-cells and Natural Killer (NK) cells (Ohta et al, 2012; Raskovalova et al, 2006). These findings suggest that A2AR antagonism may be beneficial in viral diseases, by boosting CD8 + T-cells and helping the adaptive immune response to eradicate the infection.

Adenosine analogs are drugs that resemble the chemical structure of adenosine and are used as antivirals, antibiotics, and/or antineoplastic drugs. REM, the first FDA-approved antiviral drug for COVID-19, is a pro-drug that is rapidly converted in the body to its active metabolite GS-441524 (GS), an adenosine analog, which is the form with the longest biological half-life and best tissue distribution, as observed in mice and COVID-19 patients treated with REM (Hu et al, 2021; Leegwater et al, 2022). Adenosine analogs, like GS, have a known intrinsic antiviral effect through the blockade of viral replication (Kokic et al, 2021). However, given the similarities between adenosine analogs and adenosine, we hypothesized that these compounds may have immunomodulatory properties, distinct from their intrinsic antiviral effect, through interaction with the adenosine receptor A2AR. Consistently, cases have been reported in which a rebound in lymphocyte counts has been observed in COVID-19 patients following treatment with REM (Buckland et al, 2020), a phenomenon which cannot be solely explained by REM antiviral activity.

In this study, we report that adenosine analogs predicted to bind to A2AR have previously unknown immunomodulatory properties which contribute to their antiviral activity. Specifically, we found that, GS, the metabolite of REM, acts as an A2AR antagonist, a function which is distinct from its intrinsic antiviral activity. These findings support a new rationale for the design of next-generation antiviral therapeutics with dual—immunomodulatory and intrinsic—antiviral properties. These compounds could represent game-changing therapies against emerging viral diseases and future pandemics.

# Results

## Molecular docking of adenosine analog drugs predicts interaction with A2AR

Given the striking structural similarities between the A2AR natural ligand, adenosine, and adenosine analog drugs, we hypothesized that these compounds may interact with A2AR and exert immunomodulatory effects in addition to their intrinsic antiviral activity. To test this idea, we used computational molecular docking, to predict the interaction between adenosine analogs and A2AR. To validate the reliability of our molecular docking analysis, we first tried to dock adenosine to A2AR and compared it to the known crystal structure of A2AR in complex with adenosine, which was resolved in (Lebon et al, 2011; Lebon et al, 2011). We found that our modeling was able to reproduce the crystal structure of the A2AR-adenosine complex (Fig. EV1A), confirming that our computational modeling can reliably predict the interaction between these molecules. We then screened several adenosine analogs that are in clinical development, or clinical use, as antivirals and/or antineoplastic agents (Table 1; Appendix Fig. S1). We found that some adenosine analogs are predicted to possess a higher or similar affinity to A2AR compared to adenosine, as indicated by the ΔG value, and with similar interacting residues as adenosine (Table 1; Appendix Fig. S1). Among all the screened analogs, GS-441524 (GS), the active metabolite of the antiviral drug remdesivir (REM), had the highest predicted affinity for A2AR and very similar interacting residues as adenosine. Furthermore, GS and adenosine shared striking structural similarities (Fig. EV1B), suggesting that GS could potentially interact with A2AR. These observations prompted us to select GS to test whether it may have immunomodulatory effects through interaction with A2AR, in addition to its direct antiviral activity. To test our hypothesis, we carried out an in-depth analysis of the binding between GS and A2AR compared to the adenosine-A2AR complex. Of the 9 predicted GS-A2AR states obtained from the molecular docking simulation (Fig. EV1C), the first two states exhibit higher calculated affinities for A2AR than adenosine itself, as indicated by the ΔG values of GS (−8.4, −8.3 kcal/mol) and adenosine (−7.7 kcal/mol) in complex with A2AR (Figs. 1A–C and EV1C). In addition, these two states display similar bonds as adenosine-A2AR (Figs. 1D and EV1C, EV2A). The first state showed a different orientation of GS compared to adenosine, while the second state has an almost identical orientation to adenosine (Fig. 1A–C). Both GS-A2AR complexes shared with adenosine-A2AR key hydrogen bonds, including the ones with SER-277, HIS-278, GLU-169, and ASN-253 (Figs. 1A–D and EV2B).

Since there are other adenosine receptors (A2BR, A1R, and A3R), apart from A2AR, and adenosine interacts with all of them, we asked if GS may interact with other adenosine receptors. We obtained docking results, using the same methodology, with A1R and A2BR crystal structures (Cai et al, 2022; Draper-Joyce et al, 2018). Docking with A3R was not attempted, because the structure of this receptor has not been resolved yet. Our modeling showed that GS has a similar affinity for A1R as adenosine and binds to the A1R active site (Fig. EV3); GS has a lower affinity than adenosine for A2BR and does not appear to bind within the active site of the receptor (Fig. EV4). Based on our molecular docking analysis, we

**Table 1. Screening of adenosine analogs for A2AR interaction using molecular docking.**

| Name | ΔG (kcal/mol) | Interacting residues | Use |
|---|---|---|---|
| Adenosine | −7.7 | Glu-169, Asn-253, Ser-277, His-278 | Physiological immunomodulatory metabolite, a ligand of the adenosine receptors |
| GS-441524 (remdesivir metabolite) | −8.4 | Glu-169, Asn-253, Ser-277, His-278 | Antiviral: remdesivir, the pro-drug of GS-441524, is FDA-approved for COVID-19 |
| Riboprine | −8.2 | His-278, Ser-277 | Antiviral: effective against SARS-CoV-2 in vitro<br>Antineoplastic: effective in melanoma mouse models<br>Others: evaluated in clinical trials for nausea and surgical site infection |
| Forodesine | −7.3 | Glu-169, Asn-253, Ser-277, Thr-88 | Antineoplastic: In clinical trials to treat acute lymphoblastic leukemia and B-lineage acute lymphoblastic leukemia<br>Antiviral: effective against SARS-CoV-2 in vitro |
| Galidesivir | −7.4 | Ser-277, Thr-88 | Antiviral: broad-spectrum antiviral activity against RNA viruses in vitro and in vivo, in clinical trials for various infectious diseases including COVID-19 |
| 8-chloroadenosine | −7.2 | His-278 | Antineoplastic: in clinical trials for chronic lymphocytic leukemia and acute myeloid leukemia |
| Maribavir | −7.2 | Tyr-9 | Antiviral: FDA-approved for cytomegalovirus infection |
| Vidarabine | −7.0 | Ser-277 | Antiviral: FDA-approved for herpes simplex virus encephalitis and herpes zoster infections. |
| Aristeromycin | −7.0 | Glu-169, Asn-253, Ser-277 | Antimicrobial: antibiotic and antiviral activity against various microbes, predicted to inhibit SARS-CoV-2 replication<br>Antineoplastic: effective in prostate cancer in vitro |
| Decoyinine | −6.4 | Ser-277 | Antimicrobial: inhibits Bacillus subtilis, predicted to inhibit SARS-CoV-2 replication<br>Antineoplastic: effective in melanoma mouse models |

predicted that GS interacts with A2AR with high affinity and may also interact, to a much lesser extent, with A1R.

## The adenosine analog GS-441524 antagonizes A2AR activation and downstream immunological effects

The molecular docking analysis was useful to predict the interaction between GS and A2AR. However, we did not know whether this interaction leads to blockade or activation of the receptor. To determine whether GS has A2AR-mediated effects, we treated mice with known A2AR agonist CGS-21680 (CGS), known A2AR antagonist ZM-241385 (ZM), or REM. We used REM for in vivo studies instead of GS because REM has a superior pharmacokinetics and ensures optimal tissue distribution of its metabolite GS (Leegwater et al, 2022; Hu et al, 2021). We observed a dose-dependent increase in the neutrophils-to-lymphocytes (NLR) ratio and a reduction in the number of splenic CD69 + CD3+ activated T-cells in CGS-treated mice (Fig. 2A), in agreement with the suppressive effect of activating A2AR in T-cells; whereas treatment with the A2AR antagonist ZM or REM did not change these readouts (Fig. 2A). A2AR antagonists, which block the A2AR, do not usually show significant effects in the absence of an A2AR agonist, similar to what we observed for the known A2AR antagonist, ZM, and REM. None of the treatments changed the weight of the mice (Appendix Fig. S2). Therefore, based on these observations, we hypothesized that REM metabolite, GS, could act as an A2AR antagonist, like ZM. To address this hypothesis, we tested whether GS can antagonize the A2AR agonist CGS in a number of in vitro and in vivo studies.

We first performed a dose-response study by treating primary mouse CD8 + T-cells, activated with anti-CD3/CD28, with increasing doses of CGS against a fixed dose of GS (Fig. 2B). In this assay, we used CD8 + T-cells proliferation as a readout (response), since we know that CGS reduces proliferation in activated CD8 + T-cells as shown in (Monticone et al, 2022) and other works (Sorrentino et al, 2019; Leone et al, 2018; Ohta et al, 2012). As expected, we observed that CGS reduces T-cell proliferation in a dose-dependent manner (Fig. 2B). Importantly, we found that GS antagonizes CGS-mediated suppression of proliferation, reduces the efficacy ($E_{max}$) and the potency of CGS (Fig. 2B). To further test whether GS binds to A2AR and has antagonistic properties, we measured the production of cyclic AMP (cAMP), the direct secondary mediator of A2AR (Huang et al, 1997), in an A2AR-expressing cell line in response to GS, ZM, and CGS, using a luciferase assay (Fig. 2C). We found that GS antagonizes and reduces cAMP production induced by CGS, similarly to the A2AR antagonist ZM (Fig. 2C). Along the same line, we also showed that GS antagonizes the overproduction of cyclic AMP (cAMP) in response to CGS, reducing it to control (vehicle) levels, as measured by ELISA in primary mouse CD8 + T-cells (Fig. 2C).

We then asked if GS could antagonize A2AR activation in vivo and whether this translates into immunomodulatory effects. We treated uninfected, immunocompetent mice with CGS alone or in combination with REM and we analyzed immunological readouts (Fig. 2D–H). We performed the experiments in uninfected mice to exclude the possibility that any observed immunomodulatory effects may be secondary to the intrinsic antiviral activity of REM. We found that REM treatment strongly reverses CGS-induced immunological effects, including reducing NLR (Fig. 2E) and the number of circulating neutrophils, while increasing splenic T-cells and NK cells (Fig. 2F). As further proof that REM treatment

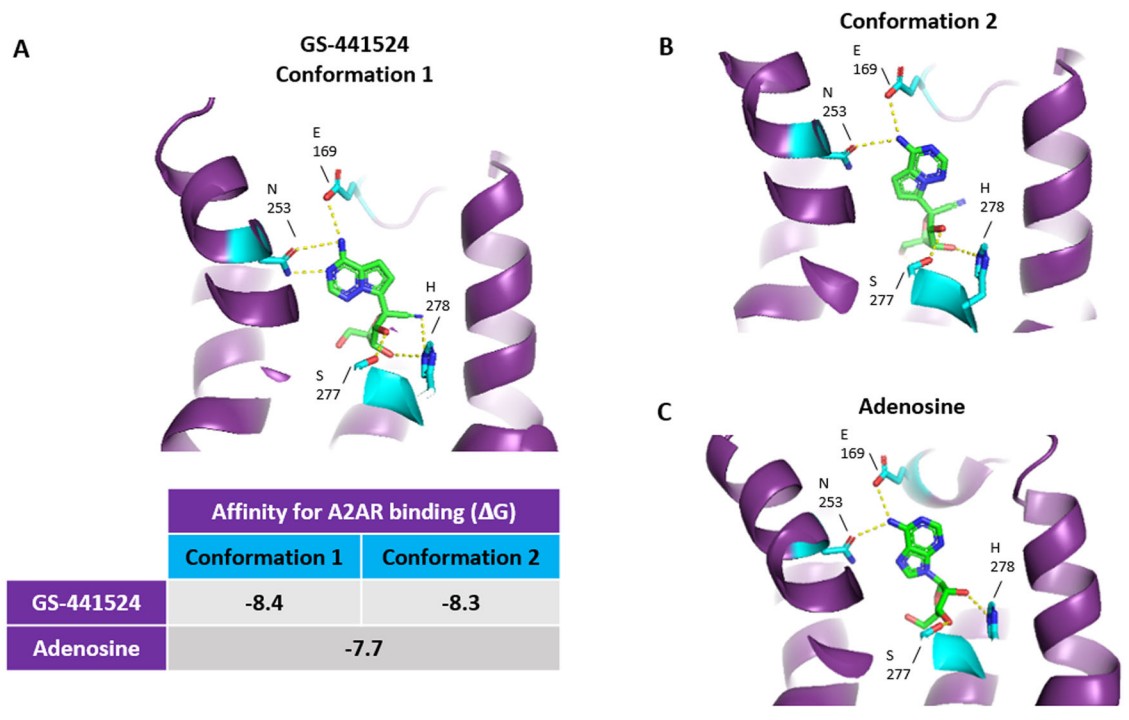

| Affinity for A2AR binding (ΔG) | | |
|---|---|---|
| | Conformation 1 | Conformation 2 |
| GS-441524 | -8.4 | -8.3 |
| Adenosine | -7.7 | |

Figure 1. Molecular docking analysis of remdesivir metabolite GS-441524 binding to A2AR.

(A, B) Computational simulation of GS (PubChem ID 44468216) binding to A2AR (PDB ID: 2YDO) using Pymol and Autodock Vina. GS-A2AR Conformation 1 (A) and Conformation 2 (B) are shown with hydrogen bonds formed with E169, N253, S277, and H278. The table shows the calculated binding affinity (ΔG, kcal/mol) for A2AR. (C) Crystallized structure of adenosine binding to A2AR (PDB ID: 2YDO). (D) Detailed presentations of adenosine and GS binding to A2AR showing the conformations from three different angles: left panels, vertical view; middle panels, 90° horizontal rotation of left panel; right panel, top view of the middle panel.

modulates A2AR, we tested whether REM treatment has any effect on NICD, the active form of Notch1, which is also known to contribute to the signaling cascade downstream of A2AR (Monticone et al, 2022). We found that REM restores NICD from CGS-mediated inhibition in CD8+ and CD4+ splenic T-cells in vivo in non-infected immunocompetent mice, as well as GS restores NICD in primary mouse CD8 + T-cells in vitro (Fig. 2G,H). In all the experiments, REM or GS did not have effects as single agents, but only when used to antagonize CGS. Overall, our results provide evidence that REM active metabolite, GS, acts as an A2AR antagonist and elicits immunological effects through A2AR blockade, independently of its intrinsic antiviral activity. We have not explored the activity of GS on other adenosine receptors, and this will be the focus of future studies.

## Treatment with a non-antiviral A2AR antagonist promotes T-cell infiltration and viral clearance in the lungs of mice infected with SARS-CoV-2

Having demonstrated an immune-stimulatory activity of GS, we tested whether stimulating CD8 + T-cell responses via A2AR blockade with immunotherapeutic A2AR antagonists could be beneficial against SARS-CoV-2 infection. We first needed to exclude that any antiviral effect observed in response to an A2AR antagonist may reflect direct antiviral activity. Therefore, we tested whether the A2AR antagonist, ZM, has any direct antiviral activity against SARS-CoV-2. We treated Vero cells with ZM or REM prior to infection with SARS-CoV-2. We did not observe a reduction in the viral burden in Vero cells in response to ZM, in contrast with REM, thus suggesting that ZM does not have intrinsic antiviral activity against SARS-CoV-2 (Fig. 3A). We then treated mice infected with SARS-CoV-2 with ZM (Fig. 3B). We observed milder weight loss, faster recovery and milder disease manifestations (clinical score) in mice receiving ZM compared to mice receiving vehicle (Fig. 3C,D). Interestingly, the histopathological analysis showed increased CD8 + T-cell infiltration in the lungs of ZM-treated mice (Fig. 3E,F) and this correlated with decreased viral load in the lungs (Fig. 3G), suggesting that ZM increases the efficiency of viral clearance by CD8 + T-cells. These findings are remarkable considering that ZM has no intrinsic antiviral activity, and suggest that systemic blockade of A2AR is beneficial for clearing SARS-CoV-2 infection, possibly by boosting CD8 + T-cell function.

It has been reported that during SARS-CoV-2 infection, IFN-γ production by CD8+ and Th1 CD4 + T-cells promotes viral clearance, whereas IL-10 and IL-17 released by Th2 and Th17 CD4 + T-cells are associated with severe disease outcomes (Huang et al, 1997; Tan et al, 2021; Pavel et al, 2021). Therefore, we asked how manipulation of A2AR changes cytokine production in different lymphocyte populations. In agreement with our previous work (Monticone et al, 2022), we found that activation of A2AR by the A2AR agonist, CGS, downregulates the active form of Notch1, NICD—a key regulator of T-cell functions, including cytokine

production (Monticone et al, 2022; Cho et al, 2009; Palaga et al, 2003)—and suppresses proliferation, whereas treatment with ZM rescues NICD and proliferation in primary mouse CD8 + T-cells and CD4 + T-cells (Fig. 4A–D). Consistently, we found that CGS and ZM antagonize each other's effects on cytokine production: CGS suppresses IFN-γ, whereas ZM rescues IFN-γ production from CGS in CD8 + T-cells and Th1 CD4 + T-cells (Fig. 4E,F); CGS does not change IL-10 in Th2 CD4 + T-cells, while ZM reduces IL-10 when used as a single treatment or against CGS (Fig. 4G); CGS increases IL-17 in Th17 CD4 + T-cells, whereas ZM restores it to control (vehicle) level, antagonizing CGS (Fig. 4G,H). Overall, our results indicate that A2AR antagonism could stimulate protective adaptive immune responses, and this could promote the clearance of SARS-CoV-2 infection.

## Remdesivir has immune-modulatory effects analogous to A2AR antagonist ZM in promoting antiviral immune responses in SARS-CoV-2-infected mice

Having found previously unknown immunomodulatory properties of REM, through the adenosine analog GS, and that a non-antiviral A2AR antagonist promotes clearing of SARS-CoV-2 in mice, we then asked if REM has similar immunomodulatory effects in SARS-CoV-2-infected animals. To test our hypothesis, we analyzed immunological readouts in mice infected with SARS-CoV-2 and treated with REM or vehicle (Fig. 5A). As expected, we observed faster recovery in mouse weight (Fig. 5B), faster viral clearance as measured by RT-PCR and IHC and milder disease manifestations (Figs. 5C and EV5A,B) in mice receiving REM compared to mice receiving vehicle. REM-treated mice started to regain weight at day 4 post-infection (Fig. 5B) and showed immune infiltration in the lungs with no significant damage (Fig. 5D), whereas we observed increased inflammation, hemorrhage, and significant damage, from bronchiolar shedding, in the lungs of mice receiving vehicle (Fig. 5D). Importantly, we found that REM treatment results in a remarkable increase of CD8 + T-cells in the lungs of infected mice (Fig. 5E,I), similar to what was observed with ZM treatment. Consistently, we found transcriptional upregulation of genes involved in adaptive immune responses and A2AR antagonism, including IFN-γ response and IL-2 signaling (Fig. 5H), in the lungs of mice treated with REM. These findings may suggest that the treatment is promoting T-cell immune responses in the lungs which contribute to viral clearance. Interestingly, we also observed a significant reduction of neutrophils (Figs. 5F and EV5C) and a drastic reduction of Neutrophils Elastase (NE) in the lungs of REM-treated mice (Figs. 5G and EV5C), which may indicate suppression of neutrophil recruitment and neutrophil extracellular traps secretion (NET) by neutrophils, key factors contributing to severe outcomes in COVID-19 (Barnes et al, 2020). In agreement with the observed increase in CD8 + T-cells and decrease in neutrophils in the lungs, we also observed the remarkable reversal of NLR in

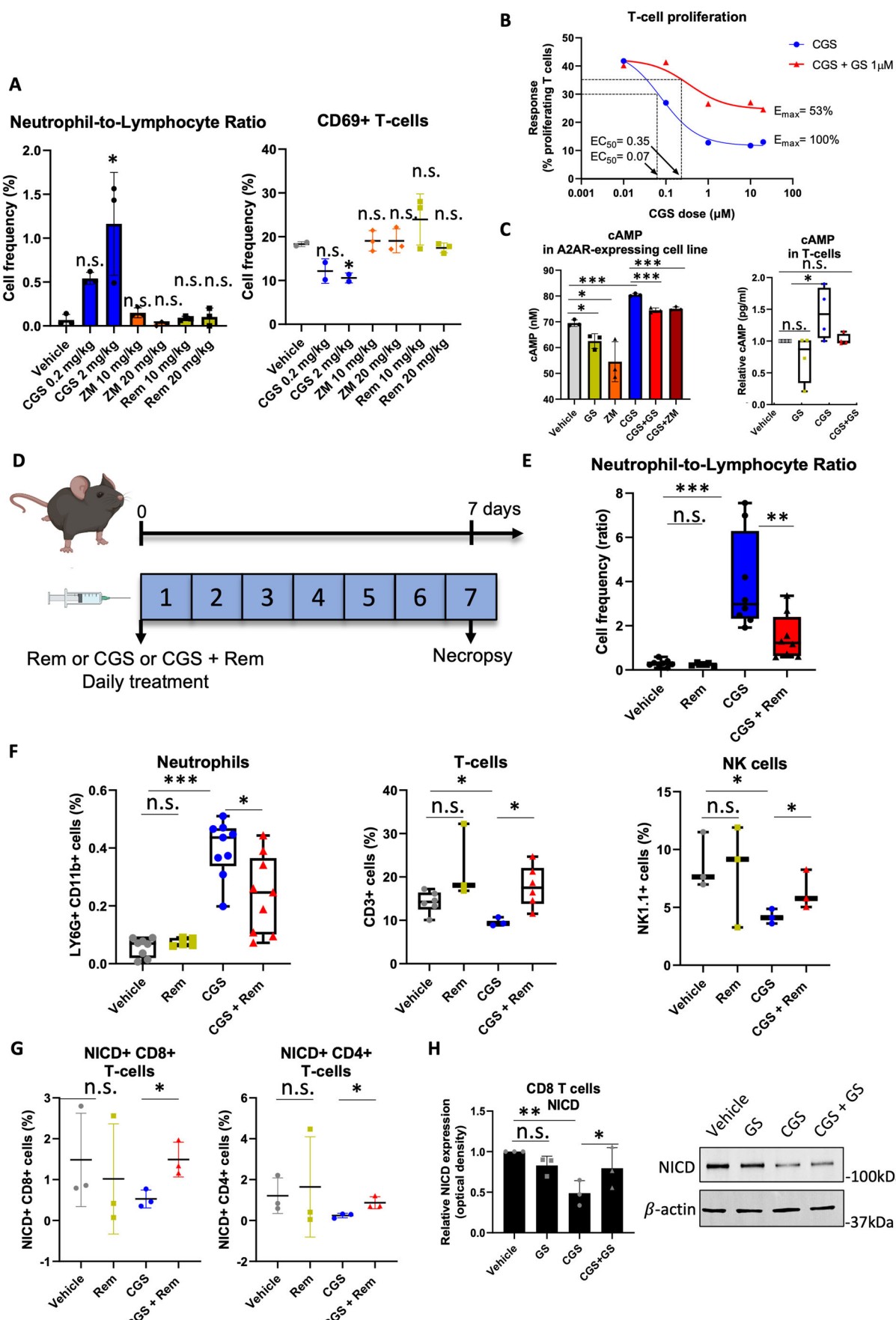

**Figure 2.   The remdesivir metabolite GS-441524 counteracts the immunological effects of an A2AR agonist.**

(A) Left panel: neutrophil-to-lymphocyte ratio (NLR) measured by flow cytometry of lymphocytes and neutrophils in the blood of C57BL/6 mice ($n = 3$ mice per group) after 7-days treatment with different doses of CGS, ZM, and REM. Right panel: frequency of splenic CD69$^+$ CD8$^+$ T-cells in C57BL/6 mice ($n = 3$ mice per group) after 7-days treatment with different doses of CGS, ZM, and REM. (B) Dose-response curve of proliferation in anti-CD3/28-activated mouse CD8$^+$ T-cells treated with increasing doses of CGS alone or in combination with a fixed dose of GS (1 uM). (C) Left panel: cAMP production measured by luciferase reporter assay in an A2AR-expressing cell line treated with 0.1 μM CGS or 1 μM GS or 1 μM ZM or CGS + GS and CGS + ZM combinations ($n = 3$ cultures). Right panel: cAMP production measured by ELISA in anti-CD3/28-activated mouse CD8$^+$ T-cells after treatment with 1 μM CGS or 1 μM GS or a combination of both ($n = 4$ cultures). (D) Diagram of the experimental timeline. C57BL/6 mice were treated with 10 mg/kg REM or 2 mg/kg CGS or a combination of 2 mg/kg CGS and 10 mg/kg REM through intraperitoneal injection daily for 7 days. Mice were sacrificed on day 7. Blood and spleen were collected for further analysis. (E) NLR determined by flow cytometry analysis of lymphocytes and neutrophils in mouse blood, after 7 days of treatment with REM ($n = 5$ mice) or CGS ($n = 8$ mice) or a combination of CGS and REM ($n = 8$ mice) vs. vehicle ($n = 8$ mice). Frequencies of neutrophils and CD8$^+$ T-cells were analyzed by flow cytometry and NLR was calculated. (F) Left panel: frequency of neutrophils in mouse blood after treatment with REM ($n = 5$ mice) or CGS ($n = 9$ mice) or combination of CGS and REM ($n = 9$ mice) vs. vehicle ($n = 5$ mice); middle panel: frequencies of splenic T-cells after treatment with REM ($n = 3$ mice) or CGS ($n = 3$ mice) or combination of CGS and REM ($n = 6$ mice) vs. vehicle ($n = 6$ mice); right panel: frequencies of NK cells ($n = 3$ mice) after treatment with REM or CGS or combination of CGS and REM vs. vehicle. (G) Frequencies of Notch1$^+$ CD8$^+$ (left) and CD4$^+$ (right) T-cells in mouse spleens ($n = 3$ mice). Cell frequency was measured by flow cytometry of blood or spleen samples from mice treated with CGS or a combination of CGS and REM. (H) Right panel: Western blot of Notch intracellular domain (NICD) from mouse primary CD8$^+$ T-cells after treatment with 1 μM CGS or 1 μM GS or a combination of both. Left panel: Normalized band intensity (optical density) was quantified and normalized with β-actin ($n = 3$ cultures/blots). Data information: "$n$" indicates biological replicates. The box plots (C, E, F) show minima, maxima, mean, 75 and 25 percentiles. Data were presented as mean ± SD. Statistical significance was calculated by one-way ANOVA with Bonferroni correction. *$P \leq 0.05$, **$P \leq 0.01$, ***$P \leq 0.001$., ns non-significant. Source data are available online for this figure.

response to REM (Fig. 5J), which was also accompanied by the reduction of neutrophils and an increase in CD3 + T-cells in circulation (Fig. EV5D,E). These effects are in agreement with our results in uninfected mice treated with REM or infected mice treated with ZM, thus suggesting that these effects are largely independent of REM intrinsic antiviral activity and dependent on its immunomodulatory properties. Furthermore, these immunological effects were observed at 4-days post-infection when mice treated with REM started to regain weight, whereas mice treated with vehicle experienced the worst decrease in weight (Fig. 5B), suggesting that the immunological effects of REM may contribute to ameliorating disease manifestations and outcomes. Finally, we found decreased cAMP expression (the second mediator of A2AR) by immunohistochemistry in correspondence to CD8 + T-cell infiltration in the lungs of REM-treated mice (Fig. 6A), which supports the hypothesis that REM elicits immunological effects through A2AR antagonism.

Lastly, we used transcriptomics of lungs at 4 and 7 days post-infection to explore the effects observed on the functional modulation of immune cells and responses elicited by REM. Using KEGG pathway analysis, we found that REM significantly changes genes of pathways associated with A2AR, including the cAMP signaling pathway, purine metabolism, and HIF-1 signaling pathway (Huang et al, 1997; Poth et al, 2013) (Fig. 6B), thus further suggesting that REM acts on A2AR. In agreement with the different disease course observed in REM-treated mice vs. vehicle, we identified a significant number of differently expressed genes (DEGs) when comparing the lungs of REM-treated mice vs. vehicle at 4 and 7 days post-infection (Fig. 6C). To interrogate the function of the identified DEGs we performed Gene Ontology (GO) analysis of Biological Processes (BP). These analyses showed that REM significantly changes the expression of genes involved in immune processes at 4-days post-infection, including defense response to other organisms, immune response, T-cell chemotaxis, and cytokine secretion (Fig. 6D), and genes involved in tissue repair processes at 7-days post-infection, including cellular component organization and biogenesis, biological adhesion, and tissue morphogenesis (Fig. 6E). Consistently, we found upregulation of pathways associated with tissue damage in COVID-19, including

Reactive Oxygen Species (ROS) pathway, EF2 targets, G2M checkpoint, mitotic spindle, and MYC targets (Ahern et al, 2022), in lung samples from mice receiving vehicle, both at 4 and 7 days post-infection (Fig. EV5F). These results are consistent with the hypothesis that REM elicits protective antiviral immune responses that lead to faster recovery and milder disease course. Collectively, our findings support the hypothesis that REM may have immunomodulatory effects which are achieved, in a significant part, through A2AR antagonism, and may be independent of its intrinsic antiviral activity.

## Discussion

In this study, we explored whether antiviral adenosine analogs are endowed with immunomodulatory properties through A2AR antagonism and how these properties contribute to their antiviral activity. We used molecular docking to screen adenosine analogs for binding to adenosine receptors. We found that several agents, including GS, the active form of the antiviral drug REM, can dock into the structure of A2AR with highly favorable ΔGs. We show that GS has A2AR antagonist-immunomodulatory properties irrespective of viral infection. These properties were evident in SARS-CoV-2 uninfected and infected mice. We show that treatment with non-antiviral A2AR antagonist ZM, results in rapid viral clearance and recovery in SARS-CoV-2-infected mice by boosting CD8+ effector function and other immune components. Importantly, our results show that GS modulates several immunological factors which are critical in the context of COVID-19: (i) stimulates T-cells: rebound in T-cell counts and function correlates with recovery and good prognosis in COVID-19 patients (Diao et al, 2020; He et al, 2020; Liao et al, 2020; Zheng et al, 2020); (ii) reduces the NLR: patients with COVID-19 were found to have an elevated NRL, and this correlates with severe disease (Liu et al, 2020a; Liu et al, 2020b); (iii) reduces neutrophils: increased neutrophil counts and NET formation have been linked with severe lung damage and Acute Respiratory Distress Syndrome (ARDS) in COVID-19 patients (Barnes et al, 2020). Therefore, our findings provide proof of principle that the immunomodulatory

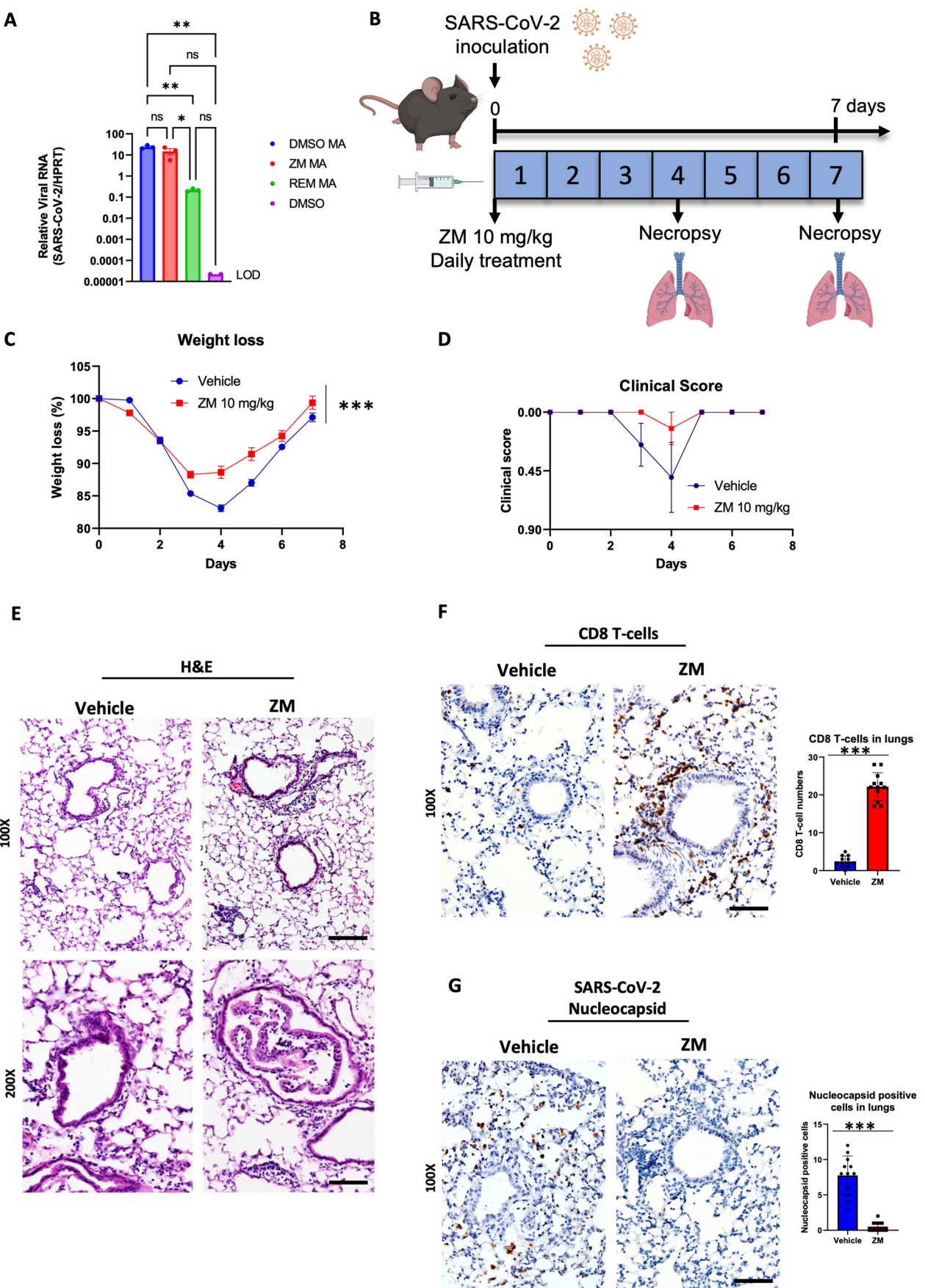

◄ **Figure 3.  Treatment with an A2AR antagonist promotes T-cell infiltration and viral clearance in the lungs of mice infected with SARS-CoV-2.**

(A) qRT-PCR titers of supernatants from SARS-CoV-2-infected Vero cells treated with 5 μM REM or 5 μM ZM vs. vehicle vs. a non-infected vehicle control ($n = 3$ cultures per treatment). (B) Diagram of the experimental timeline. C57BL/6 mice ($n = 8$ mice for each group) were infected with $10^5$ PFU SARS-CoV-2 through intranasal inoculation on day 0. The A2AR antagonist ZM (10 mg/kg) or vehicle were intraperitoneally administered to mice daily for 7 days. Mice were sacrificed on day 4 and day 7 and lungs were subject to analysis. (C) Body weight change in mice for each group over time ($n = 8$ mice for each group). (D) Clinical scores for each group over time ($n = 8$ mice for each group). (E) H&E staining, (F) IHC and quantification of CD8$^+$ T-cells ($n = 12$ image fields), and (G) IHC and quantification of SARS-CoV-2 nucleocapsid positive cells ($n = 12$ image fields) in mouse lungs at day 4. Data information: "$n$" indicates biological replicates. Data were presented as mean ± SD. Scale bars = 50 and 100 μm for 200x and 100x microscopic images, respectively. Statistical significance in qRT-PCR and IHC quantifications was calculated by two-tailed unpaired $t$-test and in mouse weight by non-linear regression. ***$P \leq 0.001$., ns non-significant. Source data are available online for this figure.

properties of a dual-function adenosine analog compound, like REM, significantly contribute to its therapeutic activity against viral infection.

REM played an important role at the beginning of the pandemic, when no evidence-based treatment was available, and still remains the main antiviral used for COVID-19, with Paxlovid. Based on our findings, REM efficacy could be explained, at least in part, by its immunomodulatory properties. Our work could explain why a rebound of T-cells was observed in some COVID-19 patients treated with REM (Buckland et al, 2020). Readouts like T-cell function and sub-population counts are not routinely measured in clinical settings in patients with infectious diseases, but our data indicate that these readouts may be important factors in predicting clinical outcomes in response to drugs like REM. Treatment with Paxlovid, which acts through a different mechanism and has a different structure than REM (Hashemian et al, 2023), was reported to lead to rebound infection with sometimes worsening symptoms in COVID-19 patients (Petrosillo, 2023). Cases of rebound infection have not been reported for REM-treated patients to date. Based on our data, it is possible that REM, by stimulating T-cell responses, prevents rebound infection from residual viruses lingering in the body after antiviral treatment. According to published clinical trials, REM is more effective in patients that did not receive oxygen therapy or received conventional oxygen therapy (Amstutz et al, 2023), when stimulating antiviral T-cells immune responses could be more beneficial, in line with our findings. Cases that require mechanical ventilation have severe inflammatory damage, and respond to immunosuppressive drugs, like dexamethasone (TheRECOVERYGroup, 2020).

Our work provides for the first time compelling evidence that it is possible to generate nucleoside analogs with dual—immunomodulatory and intrinsic—antiviral functions. Optimizing compounds for such dual activity could represent a new, superior solution for the treatment of viral infections, especially for viral diseases that impair the immune system. Stimulating long-lasting immunity could be a major therapeutic advantage, especially for viruses with low immunogenicity, like coronaviruses. Reducing viral load while stimulating T-cell responses which could potentially mimic vaccination or improve its protective efficacy by promoting long-lasting protective immunity. Adenosine signaling antagonists have been proposed as adjuvants to cancer immunotherapy (Monticone et al, 2022; Sorrentino et al, 2019; Leone et al, 2018; Willingham et al, 2018). Our results indicate that stimulating antiviral immunity is a promising application for these agents, and that dual-activity adenosine analogs with antiviral and immune-stimulatory activity can be designed.

Our study also presents some limitations that should be addressed in future studies. Our modeling was predictive about GS binding to A2AR, however it does not provide a definitive answer on the conformation of GS in complex with A2AR. Our

model suggests that GS may bind A2AR with a slightly different torsion of the molecule compared to adenosine, which could explain the difference in function of the two molecules and this will have to be confirmed using crystallography. Our data support the idea that REM has immunomodulatory effects mainly through A2AR antagonism of its metabolite GS. However, it is possible that part of the immunological effects observed in infected mice treated with REM may be a result of decreasing viral load, thus decreasing the immune deregulation induced by the infection. Studies in mice treated with ZM, a drug with no intrinsic antiviral activity, compared with mice treated with REM could be performed to better define what immunological effects are due to REM intrinsic antiviral activity versus immunomodulatory properties. In this study we provided evidence that GS acts on A2AR; however, we have not yet tested whether GS may have effects through binding to other adenosine receptors. The selectivity and pharmacology characterization of GS and other analogs should be further explored, using genetic methods and in vivo in mice. These questions need to be addressed in the future.

Our study focused on REM, however, there are numerous existing adenosine analogs that could be screened for immunomodulatory properties. Future studies will also focus on designing new analogs with improved characteristics, starting from REM and other screened dual-function analogs. Our proposed rationale could be applied in principle to any nucleoside analog antimicrobial drug. Even for anti-microbials devoid of immunomodulatory activity, the notion of combining anti-microbials with small-molecule immunostimulating adenosine antagonists could be extended to virtually any infectious disease.

In conclusion, our work introduces a new paradigm in the treatment of viral diseases, namely, the use of antivirals with dual—immunotherapeutic and intrinsic—antiviral properties. These compounds could represent game-changing therapies to control emerging infectious diseases and future pandemics.

## Methods

### Mice and biosafety

Twelve-week-old female C57Bl/6 J mice were purchased from The Jackson Laboratory. SARS-CoV-2-infected mice were housed in the Animal Biosafety Level 3 Facility at the University of Pennsylvania under the standard dark/light cycle, ambient temperature, humidity, and within the Allentown individual ventilated BCU2 caging system. Non-infected mice were housed in the LSUHSC Animal facility and kept in specific pathogen-free conditions under

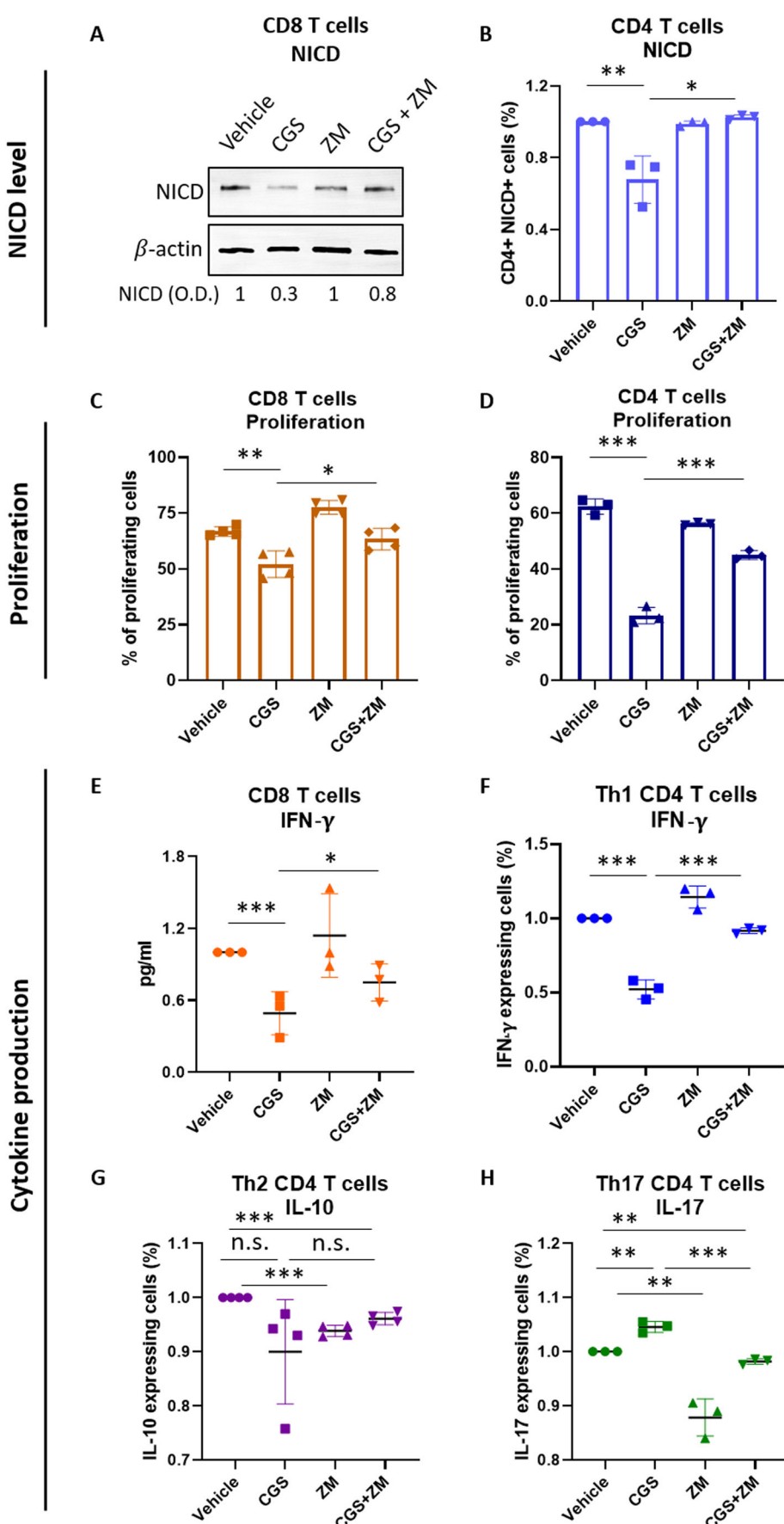

◄  **Figure 4.  Treatment with an A2AR antagonist stimulates T-cell functions.**

(A) Western blot of Notch intracellular domain (NICD) in mouse CD8+ T-cells treated with the A2AR agonist CGS or A2AR antagonist ZM or a combination of both. Band intensity (O. D.) was quantified and normalized with β-actin. A representative blot is shown ($n = 3$ blots/cultures). (B) NICD-positive CD4+ T-cells were measured using flow cytometry after treatment with CGS or ZM or a combination of both ($n = 3$ cultures). (C) Mouse CD8+ T-cell ($n = 4$ cultures) and (D) CD4+ T-cell ($n = 3$ cultures) proliferation measurement after treatment with CGS or ZM or a combination of both. The percentage of proliferating cells was measured by CFSE staining and flow cytometry. (E) IFN-γ production was measured by ELISA in supernatants of mouse CD8+ T-cells treated with CGS or ZM or a combination of both ($n = 3$ cultures). (F) Flow cytometry analysis of IFN-γ-producing Th1 CD4+ T-cells ($n = 3$ cultures), (G) IL-10-producing Th2 CD4+ T-cells ($n = 4$ cultures), (H) IL-17-producing Th17 CD4+ T-cells treated with CGS or ZM or combination of both ($n = 3$ cultures). The drug concentration used is 1 μM for all treatments. All assays were performed in cultures of anti-CD3/28-activated primary immune cells isolated from uninfected C57BL/6 mice. Data information: "n" indicates biological replicates. Data were presented as mean ± SD. Statistical significance was calculated by one-way ANOVA with Bonferroni correction. *$P \leq 0.05$, **$P \leq 0.01$, ***$P \leq 0.001$., ns non-significant. Source data are available online for this figure.

standard dark/light cycles and ambient conditions. All work was in adherence to the University of Pennsylvania's and Louisiana Health Sciences Center—New Orleans (LSUHSC-NO) approved ABSL-3 IBC and IACUC protocols.

## Virus inoculation and drug administration

Mice were randomly assigned to one of the treatment groups: mock, ZM (ZM-241385, Tocris) or REM (Remdesivir, Tocris). Mice were intraperitoneally injected with 10 mg/kg ZM or REM, 2 h prior to infection and the treatment was continued daily for the duration of the experiment. Two-hours post-treatment, mice were anesthetized with ketamine-xylazine and infected with $10^5$ PFU's of mouse-adapted SARS-CoV-2 (MA30) (Dinnon et al, 2020) intranasally in 40 uL DMEM. The virus was propagated from Stanley Perlman's MA30 stock virus in TMPRSS2 expressing Vero cells. The TMPRSS2 Vero cells were grown in Dulbecco's modified Eagle's medium DMEM, supplemented with 10% FBS. The virus was sequenced after propagation and found to match the input strain. Non-infected mice received 0.2 or 2 mg/kg CGS (CGS-21680, Tocris) or, 10 or 20 mg/kg ZM or REM intraperitoneally daily (100 ul in PBS-DMSO) for the duration of the experiment.

## Clinical score

Mice were monitored and scored for clinical disease and body weight daily. Numerical clinical scores were assigned based on disease manifestations and weight loss rate as shown in Fig. EV5B.

## Tissue harvest

At the clinical or experimental endpoint (Day 4 or 7 post-infection), mice were anesthetized by isoflurane and necropsied. Blood was collected through cardiac puncture and placed in Sarstedt EDTA tubes and allowed to sit for 10 min. Mice were perfused transcardially with 1X PBS. Followed by lungs and spleen collected and placed in 10% neutral buffered formalin separately. All tissues were inactivated and removed from the Animal Biosafety Level 3 facility in accordance with the University of Pennsylvania IBC protocol.

## In vitro infection and viral load via RT-qPCR

Vero cells were maintained in Dulbecco's modified Eagle's medium (DMEM), supplemented with 10% FBS and 1% Pen Strep and plated in six-well tissue culture plates at 1-million cells per well. Cells were allowed to adhere overnight in a $CO_2$ incubator at 37 °C. The following day, the culture media was aspirated from each well

and 2 mL of fresh DMEM + drug were added. After an hour of incubation, cells were infected with a multiplicity of infection (MOI) of 0.05 in DMEM + drug. Two hours later, the virus was removed and washed with 1X phosphate-buffered saline (PBS). Fresh DMEM + drug was added to mock or virus-exposed cells and incubated for 24 h. Twenty-four hours post viral exposure, the cellular layer within each well was collected in TRIzol and homogenized. The cellular homogenate was added to Trizol at a 1:3 ratio and extracted using a Zymo Direct-zol RNA kit following the manufacturer's protocol. RNA was reverse transcribed and amplified using iScript cDNA Synthesis Kit (Biorad). Gene-specific primers for SARS-CoV-2 N gene (F:TTACAAACATTGGCCG-CAAA and R:GCGCGACATTCCGAAGAA) and human HPRT1 (F: CATTATGCTGAGGATTTGGAAAGG and R: CTTGAGCA-CACAGAGGGCTACA) with Power SYBR Green PCR Master Mix (Applied Biosystems) were used to amplify viral and cellular RNA by QuantStudio 3 Flex Real-Time PCR Systems (Applied Biosystems). The relative expression levels of target genes were calculated using HPRT1 RNA as an internal control.

## Isolation and culture of primary immune cells

CD8+ and CD4 + T-cells were aseptically isolated from the spleens and lymph nodes of C57BL/6 (6–12 weeks old) mice using the negative selection Easysep mouse CD8 + T-cell isolation kit or Easysep mouse CD4 + T-cell isolation kit (StemCell Technologies) according to the manufacturer's instructions. CD8+ and CD4 + T-cells were cultured in RPMI supplemented with 10% fetal bovine serum, 4 mM L-Glutamine, 50 U/ml penicillin, 50 μg streptomycin, and 50 μM 2-mercaptoethanol. The cells were activated in plates coated with anti-mouse CD3ε and anti-mouse CD28 antibodies (1 μg/ml, BD Biosciences) for up to 72 h. Th1, Th2, and Th17 CD4 + T-cells were differentiated and cultured using the CellXVivo Mouse Th1 Cell differentiation Kit, CellXVivo Mouse Th2 Cell differentiation Kit, or the CellXVivo Mouse Th17 Cell differentiation Kit (Biotechne, R&D Systems) according to the manufacturer's instructions. T-cell proliferation was measured by labeling primary CD8+ or CD4 + T-cells with 1 μM carboxyfluorescein diacetate succinimidyl ester (CFSE, Thermo Fisher) for 10 min at 37 °C before activation. T-cell proliferation was measured by flow cytometry (Beckman Coulter) as indicated by CFSE dilution, and data were analyzed using Kaluza software (Beckman Coulter). Neutrophils and macrophages were isolated from the peritoneal cavity of C57BL/6 (6–12 weeks old) mice at 24 and 96 h post-injection with 3% Brewer thioglycolate medium (Zhang et al, 2008). After euthanizing the mice, the collection of peritoneal cells was performed by injecting 3 ml of 1X PBS into the peritoneal cavity and carefully aspirating the fluid at least 5 times.

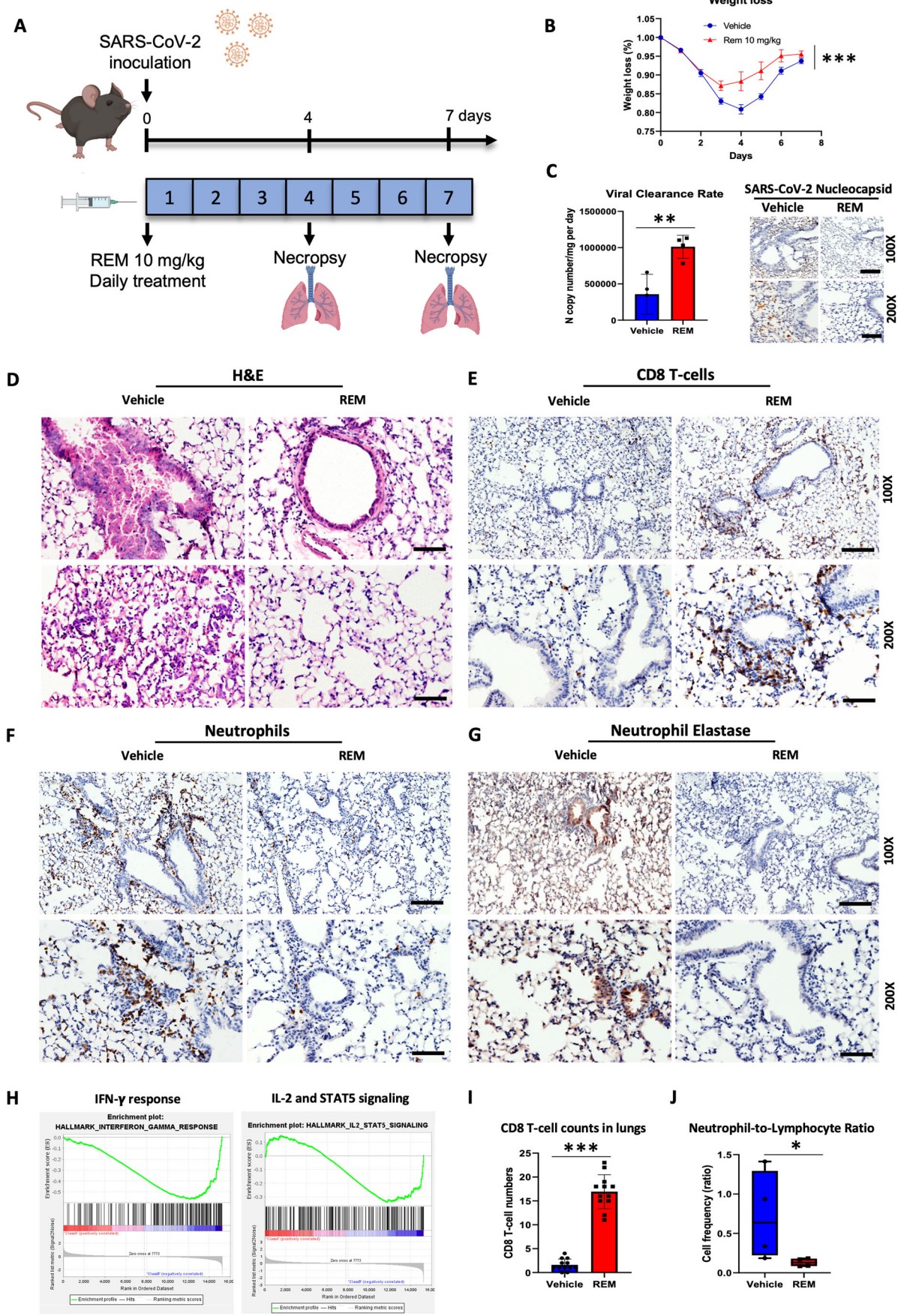

**Figure 5. Remdesivir promotes antiviral immune responses which contribute to the clearance of SARS-CoV-2.**

(A) Diagram of the experimental timeline. C57BL/6 mice ($n = 8$ mice for each group) were infected with $10^5$ pfu SARS-CoV-2 through intranasal inoculation on day 0. REM (10 mg/kg) or vehicle was administered to mice daily for 7 days. Mice were sacrificed on day 4 and day 7 and lungs were subject to analysis. (B) Mouse body weight change for each group over time ($n = 8$ mice per group). (C) Left panel: viral clearance rate based on viral copy number measured by RT-PCR in mouse lungs at different post-infection times ($n = 4$ mice). Right panel: IHC labeling of SARS-CoV-2 nucleocapsid positive cells in the lungs of mice sacrificed on day 4. (D–G) Lung sections of mice sacrificed on day 4 were stained with (D) H&E and immunolabeled for (E) CD8, (F) Ly6G, and (G) Neutrophil elastase. Representative pictures are shown. (H) GESA transcriptional analysis of IFN-γ response pathway (left) and IL-2-STAT5 pathway (right) between control and REM-treated groups sacrificed on day 4 (Class A: vehicle group; Class B: REM-treated group). (I) A number of CD8$^+$ T-cells in the lungs in vehicle and REM-treated mice were quantified in IHC lung sections ($n = 12$ image fields). (J) The neutrophil-to-lymphocyte ratio in the blood of mice sacrificed on day 4 ($n = 4$ mice). The box plot shows minima, maxima, mean, 75, and 25 percentiles. Data information: "$n$" indicates biological replicates. Data were presented as mean ± SD. Scale bars = 50 and 100 µm for 200x and 100x microscopic images, respectively. Statistical significance was calculated by a two-tailed unpaired t-test (C, I, J) or non-linear regression (B). *$P \leq 0.05$, **$P \leq 0.01$, ***$P \leq 0.001$. Source data are available online for this figure.

## Tissue processing and staining for flow cytometry

Collected mouse blood was transferred into the ACK lysis buffer and incubated at room temperature for 5 min. 1X PBS was added to stop lysis and then centrifuged at $300 \times g$ for 5 min at 4 °C. This was completed twice. The supernatant was removed, and the pellet was resuspended in 5 mL of 4% paraformaldehyde (PFA). Mouse spleens were collected, minced by gently crushing the spleen using the flat end of the 3 cc syringe plunger, and then filtered through a 70 µm filter. The cell suspension was centrifuged at $400 \times g$ for 5 min and the supernatant was removed. ACK lysing buffer (1 ml) was added and incubated at RT for 2 min. 1X PBS was added to stop lysis and centrifuged at $300 \times g$ for 5 min at 4 °C, the supernatant was removed, and the pellet was resuspended in 1X PBS. Prior to staining, Fc block (1 µg/$10^6$ cells) was added to samples and incubated for 5 min at RT. Samples were then incubated with antibodies for 30 min at 4 °C, washed with PBS, and centrifuged at $400 \times g$ for 5 min. Samples were fixed with 4% PFA for 20 min at 4 °C. For intracellular staining, 250 µl Permeablization buffer (BD Transcription Factor Perm Set) was added and incubated for 20 min at 4 °C, washed with PBS, and centrifuged at $400 \times g$ for 5 min. Samples were then Incubated with antibodies for 30 min at 4 °C. Samples were washed, resuspended in PBS and run on a flow cytometer. Antibody used in this study: anti-CD45 (BD 553079), anti-NICD (BD 552768), anti-IFN-γ (BD 562303), anti-NK1.1 (BD 566502), anti-CD4 (BD 563933), anti-CD69 (BD 560689), FVS780 (BD 565388), anti-CD3 (BD 562600), anti-CD8 (BD 563046), anti-Ly6G (BD 551461), anti-CD11b (BD 557686), anti-IL-10 (BD 554468), and anti-IL-17 (BD 559502). T-cell proliferation was measured by labeling primary CD8+ or CD4 + T-cells with 1 µM carboxyfluorescein diacetate succinimidyl ester (CFSE, Thermo Fisher) for 10 min at 37 °C before activation and treatments. Samples were run on a Gallios cytometer (Beckman Coulter) and data analyzed using Kaluza software (Beckman Coulter). A diagram of the gating strategy is provided in Appendix Fig. S3.

## Histology and immunohistochemistry

Mouse lungs were inflated and fixed in 10% buffered formalin for tissue processing. After paraffin embedding, tissues were sectioned at a thickness of 4 µm, placed on electromagnetically charged slides (Fisher Scientific; Waltham, MA), and stained with Hematoxylin & Eosin (H&E) for routine histologic analysis. Immunohistochemistry was performed using the Avidin-Biotin-Peroxidase complex system, according to the manufacturer's instructions (Vectastain Elite ABC Peroxidase Kit; Vector Laboratories, Burlingame, CA). Our modified protocol includes meting the paraffin at 56 °C for 15 min, clearing in xylenes three times for 15 min each, rehydration through descending grades of alcohol up to water, non-enzymatic antigen retrieval with

0.01 M sodium citrate buffer pH 6.0 at 95 °C for 25 min in a vacuum oven, and endogenous peroxidase quenching with 3% $H_2O_2$ in methanol, Following these steps, slides were washed with PBS and blocked in PBS/0.1% BSA containing 5% normal horse serum (for mouse monoclonal antibodies), or normal goat serum (for rabbit generated antibodies) for 2 h at room temperature, then incubated overnight with primary antibodies. Primary antibodies included a rabbit polyclonal anti-nucleocapsid of SARS-CoV-2 (Novus Biologicals, NB100-56576, 1:200 dilution), a rabbit recombinant monoclonal anti-CD8a (Abcam, EPR21769, 1:2000 dilution), a mouse monoclonal anti-cyclic adenosine monophosphate (cAMP) (AbboMax, Clone ABM486, 1:200 dilution), a rabbit polyclonal anti-Ly-6G (Invitrogen, PA5-141170, 1:250 dilution) and a rabbit recombinant monoclonal anti-Neutrophil Elastase (Invitrogen, Clone 8X4E6, 1:100 dilution). After overnight incubation, slides were rinsed thoroughly with PBS and incubated with biotinylated secondary anti-rabbit or anti-mouse antibodies for 1 h at room temperature. Then, Avidin-Biotin complexes were incubated for 1 h at room temperature, and finally, slides were developed using a diaminobenzidine substrate (DAB, Roche Laboratories), counterstained with Hematoxylin, and mounted with Permount. Images were collected using a BX61 Olympus microscope equipped with a high-resolution Olympus DP80 digital camera and CellSense image capture software.

## RNA sequencing

Total RNA was extracted from FFPE tissue samples using the AllPrep DNA/RNA FFPE kit (Qiagen). Directions from the manufacturer were followed. Tissue sections of 20 µm were first deparaffined using Sigma's xylene substitute, followed by two ethanol washes and air dried for 10 min at 37 °C. After deparaffinization, tissues were lysed in a lysis buffer containing proteinase K for 15 min at 56 °C. RNA containing supernatant was then de-crosslinked at 80 °C for 15 min and purified using the RNA purification columns as indicated by the manufacturer's instructions. A treatment with DNase I (Qiagen) was included. RNA was eluted in 20 µl of $H_2O$ after 5 min of incubation at room temperature. RNA quantification was performed using the Qubit RNA HS Assay kit (Invitrogen) and RNA quality was assessed with the Agilent 2100 bioanalyzer (Agilent Technologies). Libraries were generated using Illumina's TruSeq RNA exome library preparation kit and according to the manufacturer's instructions, with the following modifications (Wen et al, 2017): 300 ng of input RNA was used, PCR cycles were reduced to nine cycles, and hybridization time was extended to 16 h. After library preparation, free adapters were blocked using Illumina's free adapter-blocking reagent.

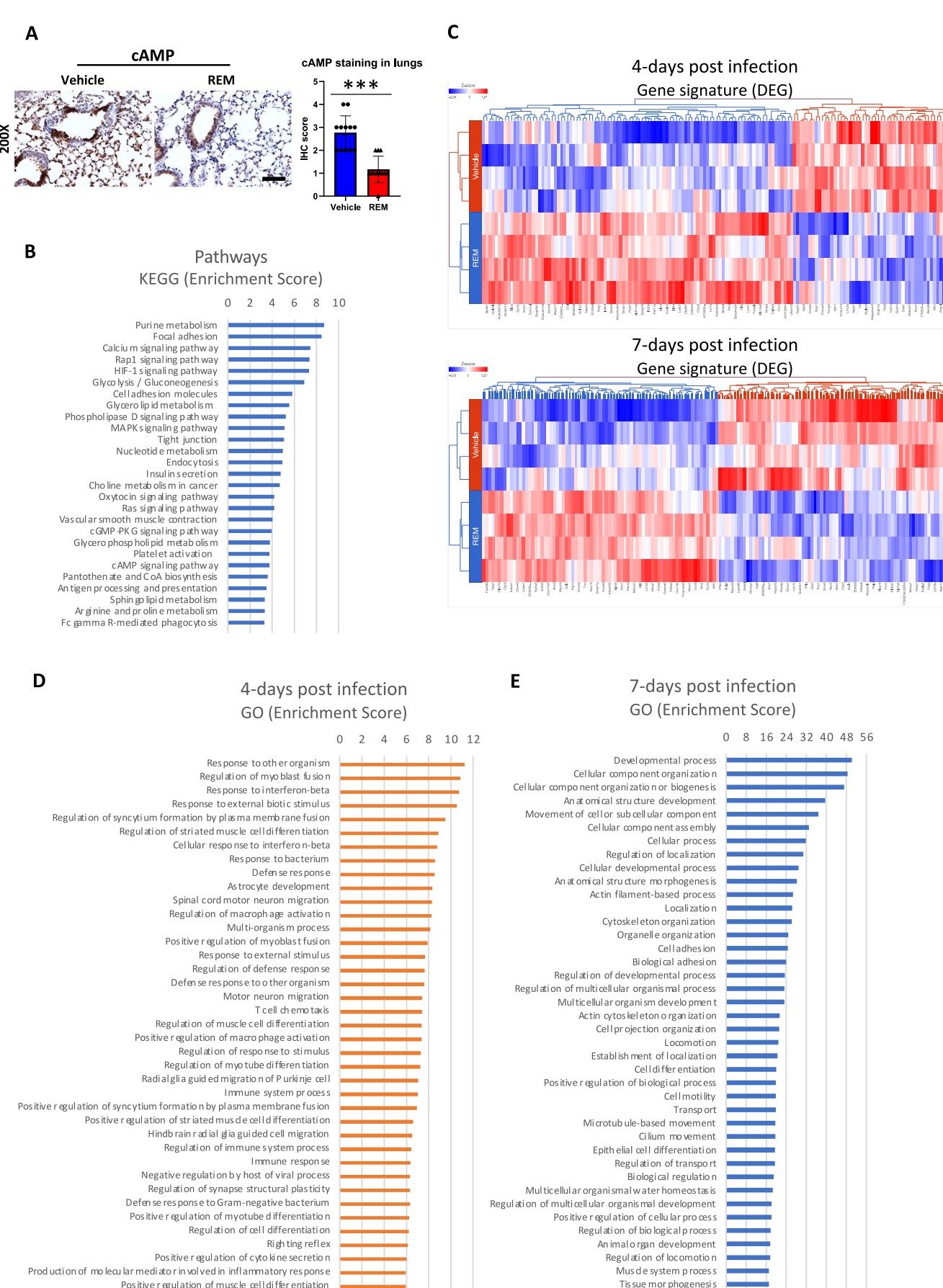

Figure 6. Gene and pathway signatures in remdesivir-treated mice reflect A2AR antagonism, immune responses, and tissue repair.

(A) IHC labeling and quantification of cAMP in the lungs of vehicle and REM-treated groups ($n = 12$ image fields). Scale bar $= 50$ μm. (B) KEGG pathway analysis of significantly changed pathways in the lungs of REM treated vs. vehicle group. (C) Heatmaps of differentially expressed genes (DEG) in the lungs of vehicle and REM-treated groups on day 4 (top) and day 7 (bottom) ($n = 4$ mice per group per time point). (D, E) Gene Ontology (GO) analysis of the top 40 significantly changed pathways in the lungs of REM treated vs. vehicle group on day 4 (left) and day 7 (right). Data information: "$n$" indicates biological replicates. Data were presented as mean ± SD. Statistical significance was calculated by a two-tailed unpaired $t$-test. ***$P \leq 0.001$. Source data are available online for this figure.

Libraries were validated on Agilent's 2100 bioanalyzer (Agilent Technologies) using a DNA 1000 kit and quantified using the Qubit dsDNA HS Assay kit (Invitrogen). Sequencing was performed on Illumina's NextSeq 2000 instrument using a NextSeq 2000 P2 200 cycles kit with pair-end 76 bp reads.

FASTQ files from the NextSeq2000 run were downloaded from the Illumina BaseSpace hub and uploaded to Partek Flow. Contaminants (rDNA, tRNA, and mtrDNA) were removed using Bowtie v2.2.5. Sequence reads were then aligned to STAR v2.7.8a to the mm10 version of the mouse genome. The reads were quantified with RefSeq Transcripts 96. For differential analysis, the samples were split based on days of treatment (4 and 7 days), and filters were applied to remove features with a maximum of five reads across all samples. The remaining features (>15,000) were normalized with TMM and transformed by log2 (+0.0001/TMM/log2). Heatmaps, pathways maps and gene ontology (GO) were all obtained by the tools embedded in Partek Flow. The sequencing data have been deposited to the Gene Expression Omnibus (GEO) under the accession number GSE236563.

## SARS-CoV-2 RT-PCR

Prepared viral RNA was assayed using the Seegene Allplex™ 2019-nCoV2 assay (RP10250X) per manufacturer's instructions, run on a BioRad CFX96 Touch, and visualized in the Seegene Viewer software. A standard curve using a genomic RNA from SARS-CoV-2/USA-WA1/2020 (ATCC; VR-1986D) was used to quantitate the number of viral genomes (copy number) present in each sample. Viral clearance rates were measured by comparing viral copy numbers at different days post-infection.

## Molecular docking

Single docking experiments of adenosine receptors (A2A: Protein Data Bank ID 2YDO, A1: Protein Data Bank ID 6D9H, A2B: Protein Data Bank ID 8HDP) to adenosine (PubChem ID 60961) and GS-441524 (PubChem ID 44468216) were performed using AutoDock Vina (Trott and Olson, 2010). AutoDock Tools was used to prepare adenosine receptor 2A and ligands for docking and defining the binding site coordinates (Forli et al, 2016). Predicted structures were visualized by PyMOL (The PyMOL Molecular Graphics System, Version 2.0 Schrödinger, LLC.) and BIOVIA Discovery Studio (BIOVIA, Dassault Systèmes, Discovery Studio, San Diego: Dassault Systèmes, 2021).

## Cytokines and chemokines measurements

IFN-γ production was measured in supernatants of primary immune cell cultures using an IFN-γ mouse ELISA kit (Invitrogen). Cytokines and chemokines from macrophage and neutrophil culture supernatants were measured using the antibody-based multiplex Proteome Profiler Mouse Cytokine Array Kit (Biotechne, R&D Systems), according to the manufacturer's instructions.

## cAMP assays

The production of cAMP was measured in primary mouse CD8 + T-cells or in an A2AR-expressing cell line (Abeomics, 14-522ACL, A2AR-expressing Cho cell line) using a cAMP competitive ELISA kit (Abcam) or a luciferase reporter assay (cAMP-Glow Assay, V1501, Promega) according to the manufacturer's protocol.

## Western blotting

Protein lysates were loaded on 4–15% SDS-PAGE gels (BioRad) and transferred to PVDF membranes (Millipore). Blots were incubated at 4ºC overnight with primary antibody diluted in Intercept blocking buffer (Licor). The next day, blots were incubated with IRDye fluorescent secondary antibodies (Licor) for fluorescence detection, for 1 h at RT. Proteins were visualized by imaging the blots on an Odyssey scanner (Licor). The following primary antibodies were used for Western blot in this study: anti-Notch1 (D1E11, Cell Signaling) for Notch1 full length and cleaved forms; anti-cleaved Notch1 Val1744 (D3B8, Cell Signaling or PA5-99448, Invitrogen-Thermo Fisher) for NICD; anti-β-actin (AC-15, SCBT).

## Statistical analysis

Statistical significance was measured using a two-tailed unpaired Student's $t$-test for comparisons between two groups and one-way ANOVA with Bonferroni correction for multiple comparisons. Non-linear regression was used for comparisons of mouse weight between two sample groups over time. $P$ values $\leq 0.05$ were considered significant. Limma-trend was used to identify genes differentially expressed between treatment (REM) vs. vehicle, for both treatment days. Genes with $p$ value $<0.05$ and fold change (FC) $>2.0$ were considered significant.

# Data availability

All sequencing data obtained in this study have been deposited to the Gene Expression Omnibus (GEO) and can be accessed under the accession number GSE236563 at https://www.ncbi.nlm.nih.gov/geo/query/acc.cgi?acc=GSE236563. All flow cytometry data have been deposited in a Dryad public dataset accessible at https://doi.org/10.5061/dryad.djh9w0w6j.

The source data of this paper are collected in the following database record: biostudies:S-SCDT-10_1038-S44319-024-00189-4.

## Peer review information

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

## Acknowledgements

This work was supported by the US Department of Defense (DOD) Discovery Award grant (W81XWH-21-1-0078 to Monticone G.) and in part by the COBRE grant (NIH GM121288). We thank the Cellular Immunology and Immune Metabolism, the Molecular Histopathology and Analytical Microscopy, and the Translational Genomics core facilities at LSU Health Sciences Center and the Louisiana Cancer Research Consortium, for their contributions to this work. We would like to thank Dr. Stephen Hatfield, Dr. Maira Di Tano, and Grace Kim for providing insightful discussions.

## Author contributions

**Giulia Monticone**: Conceptualization; Resources; Data curation; Software; Formal analysis; Supervision; Funding acquisition; Validation; Investigation; Visualization; Methodology; Writing—original draft; Project administration; Writing—review and editing. **Zhi Huang**: Data curation. **Peter Hewins**: Data curation. **Thomasina Cook**: Data curation. **Oygul Mirzalieva**: Data curation. **Brionna King**: Data curation. **Kristina Larter**: Data curation. **Taylor Miller-Ensminger**: Data curation. **Maria D Sanchez-Pino**: Data curation. **Timothy P Foster**: Data curation. **Olga V Nichols**: Data curation. **Alistair J Ramsay**: Data curation. **Samarpan Majumder**: Data curation. **Dorota Wyczechowska**: Data curation. **Darlene Tauzier**: Data curation. **Elizabeth Gravois**: Data curation. **Judy S Crabtree**: Data curation. **Jone Garai**: Data curation. **Li Li**: Data curation. **Jovanny Zabaleta**: Data curation. **Mallory T Barbier**: Data curation. **Luis Del Valle**: Data curation; Supervision; Validation; Investigation; Methodology; Project administration; Writing—review and editing. **Kellie A Jurado**: Conceptualization; Data curation; Supervision; Validation; Investigation; Methodology; Writing—original draft; Project administration; Writing—review and editing. **Lucio Miele**: Conceptualization; Data curation; Supervision; Validation; Investigation; Methodology; Writing—original draft; Project administration; Writing—review and editing.

Source data underlying figure panels in this paper may have individual authorship assigned. Where available, figure panel/source data authorship is listed in the following database record: biostudies:S-SCDT-10_1038-S44319-024-00189-4.

## Disclosure and competing interests statement

The authors declare no competing interests.

# Expanded View Figures

**Figure EV1. Structure similarities and molecular docking simulations between GS-441524 and adenosine.**

(**A**) Crystallized structure (Top) and simulated structure (bottom) of adenosine binding to A2AR. (**B**) Structure of adenosine (PubChem ID 60961) (top) and GS (PubChem ID 44468216) (bottom). (**C**) Nine simulated states of GS binding to A2AR. Calculated binding affinities (ΔG, kcal/mol) are shown in the bottom table.

                                                                                                       

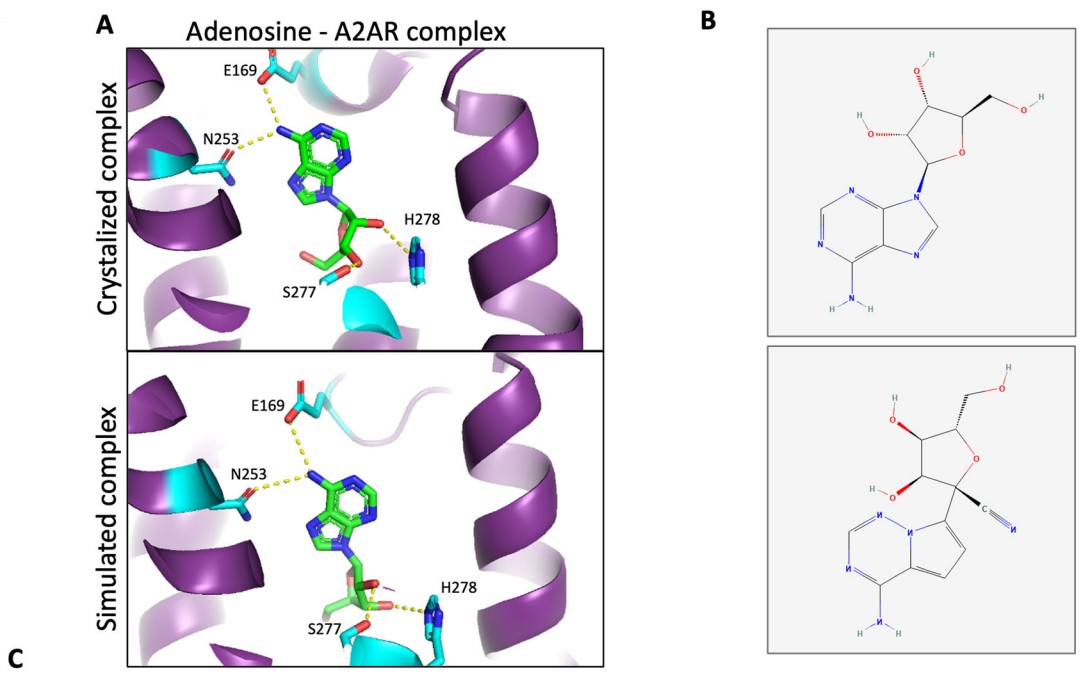

**A** Adenosine - A2AR complex

Crystalized complex

Simulated complex

**B**

Adenosine

GS-441524

**C** GS-441524 binding to A2AR

state 1

state 2

state 3

state 4

state 5

state 6

state 7

state 8

state 9

| Affinity for A2AR binding (ΔG) | | | | | | | | |
|---|---|---|---|---|---|---|---|---|
| **GS** | -8.4 | -8.3 | -8.1 | -8.1 | -7.9 | -7.8 | -7.6 | -7.5 | -7.4 |

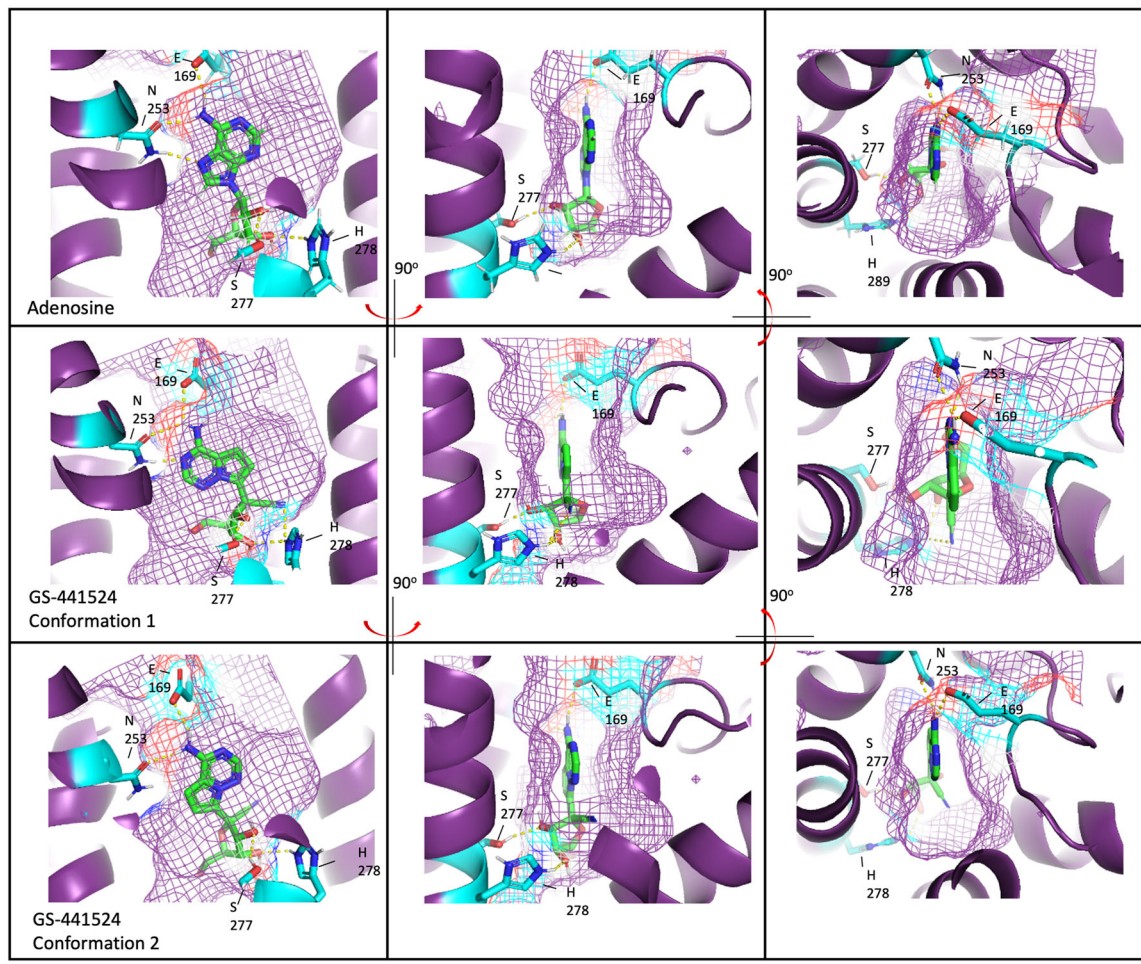

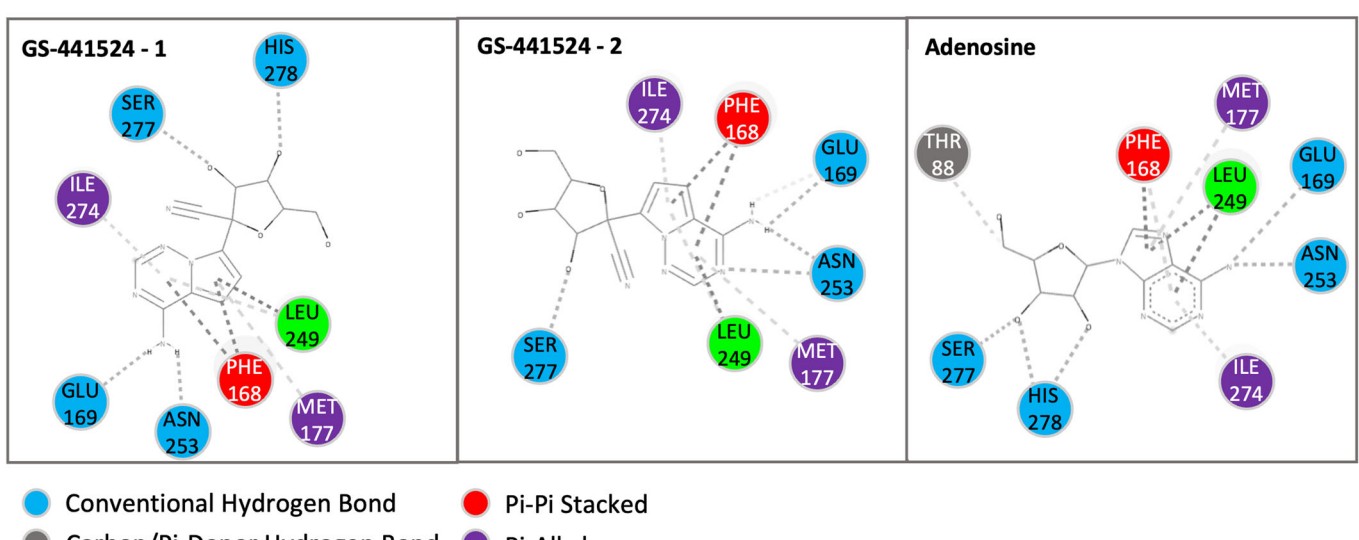

**Figure EV2. Detailed analysis of molecular bonds and interaction between GS-441524 and A2AR.**

(**A**) Detailed presentation of Fig. 3D showing the surface (mesh) of E169, N253, S277, and H278 residues in the A2AR active site in contact with GS. (**B**) 2D presentation of detailed molecular bonds and interactions of GS confirmation 1 and 2 (left and middle panels) or adenosine (right panel) with A2AR predicted using Discovery Studio software.

## A

GS-441524 binding to A1R

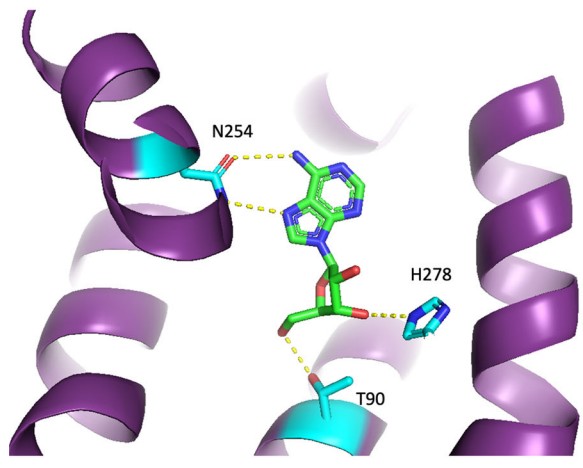

## B

## Adenosine binding to A1R

## C

### GS binding to A1R

| mode | Affinity for A1R binding (ΔG) affinity (kcal/mol) | |
| --- | --- | --- |
| | GS | adenosine |
| 1 | -6.8 | -6.7 |
| 2 | -6.6 | -6.6 |
| 3 | -6.2 | -6.6 |
| 4 | -5.6 | -6.3 |
| 5 | -5.4 | -6.2 |
| 6 | -5.3 | -6.2 |
| 7 | -5.2 | -6.1 |
| 8 | -5.2 | -6.1 |
| 9 | -5.2 | -6.1 |

◀ **Figure EV3. Molecular docking between GS-441524 and A1R.**

(**A**) Simulated binding of GS to A1R (PDB ID: 6D9H). (**B**) Simulated binding of adenosine to A1R. (**C**) Calculated binding affinities (ΔG, kcal/mol) of GS and adenosine to A1R.

**A**

GS-441524 binding to A2BR

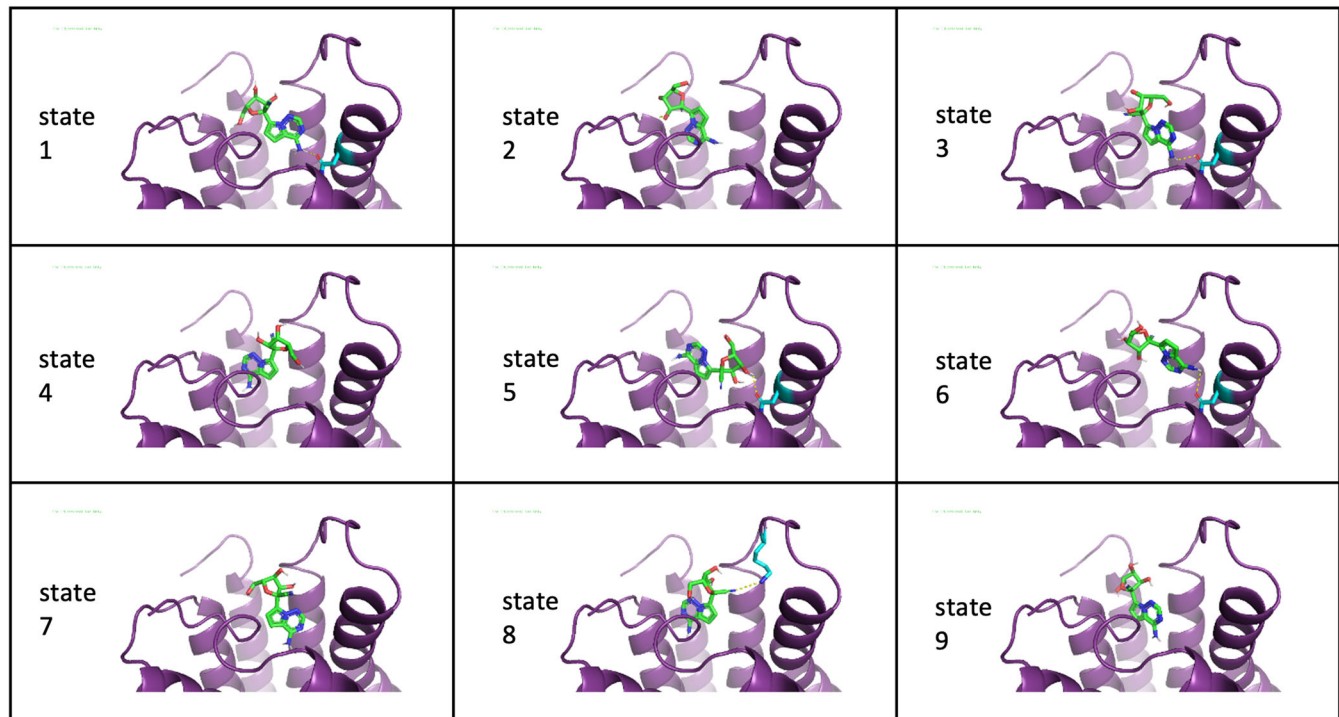

**B**

# Adenosine binding to A2BR

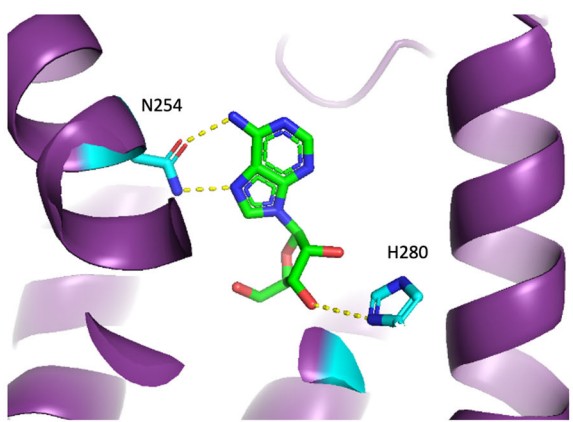

**C**

# GS binding to A2BR

| | Affinity for A2BR binding (ΔG) | |
|---|---|---|
| mode | affinity (kcal/mol) | |
| | GS | adenosine |
| 1 | -5.6 | -6.1 |
| 2 | -5.5 | -6 |
| 3 | -5.3 | -5.8 |
| 4 | -5.2 | -5.7 |
| 5 | -5.2 | -5.7 |
| 6 | -5.1 | -5.7 |
| 7 | -5.1 | -5.6 |
| 8 | -5.1 | -5.5 |
| 9 | -5 | -5.5 |

◄ **Figure EV4.   Molecular docking between GS-441524 and A2BR.**

(**A**) Simulated binding of GS to A2BR (PDB ID: 8HDP). (**B**) Simulated binding of adenosine to A2BR. (**C**) Calculated binding affinities (ΔG, kcal/mol) of GS and adenosine to A2BR.

**A**

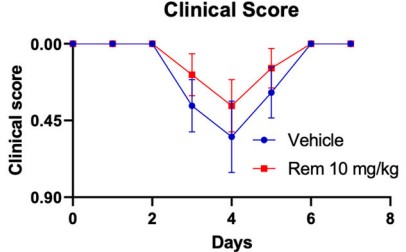

Clinical Score

**B**

| Disease Manifestations | Clinical Score |
|---|---|
| Healthy | 0 |
| Lethargic/reduced mobility | 1 |
| Change in respiration | 2 |
| Labored breathing | 3 |
| Moribund | 4 |
| Dead | 5 |

**C**

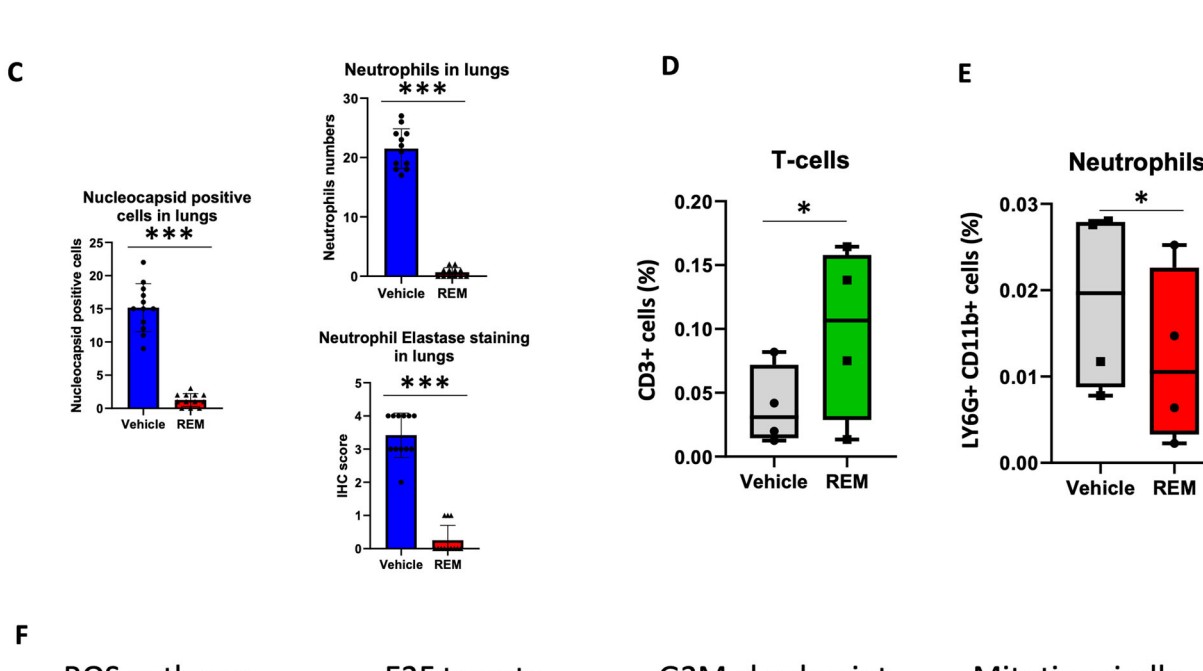

Nucleocapsid positive cells in lungs

Neutrophils in lungs

Neutrophil Elastase staining in lungs

**D**

T-cells

**E**

Neutrophils

**F**

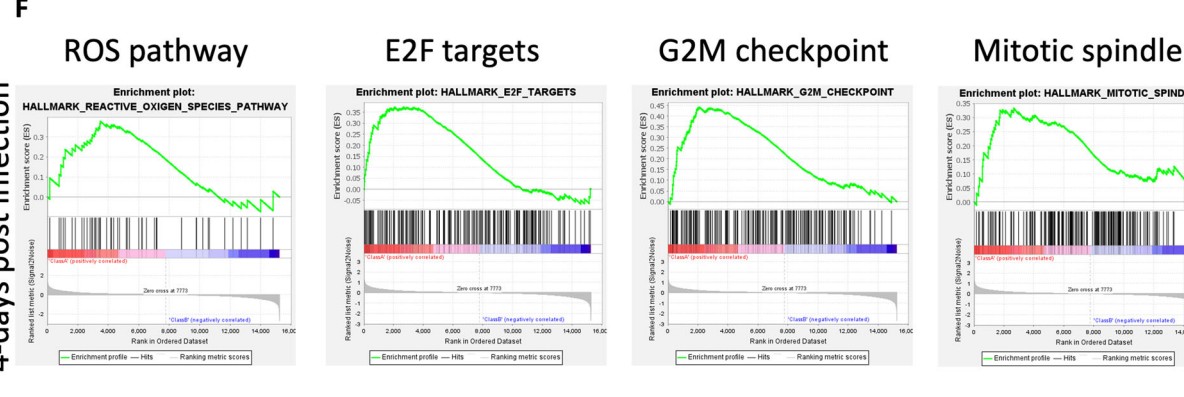

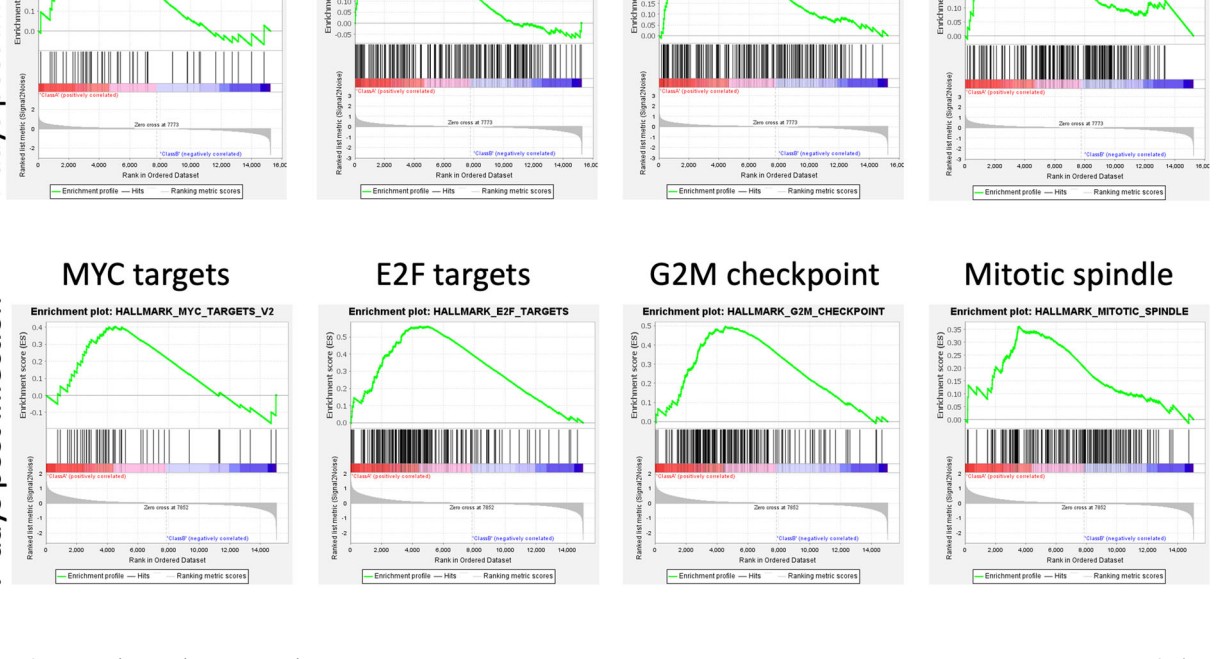

4-days post infection

ROS pathway — E2F targets — G2M checkpoint — Mitotic spindle

7-days post infection

MYC targets — E2F targets — G2M checkpoint — Mitotic spindle

◀ **Figure EV5.  Clinical and immunological readouts in SARS-CoV-2-infected mice treated with remdesivir.**

(A) Clinical score of SARS-CoV-2-infected mice treated with vehicle or 10 mg/kg REM over 7 days ($n = 8$ mice per group). (B) Table of disease manifestations and corresponding clinical score. (C) Quantification of SARS-CoV-2 nucleocapsid positive cells, neutrophils, and neutrophil elastase staining in IHC of lung sections from SARS-CoV-2-infected mice treated with REM vs. vehicle ($n = 12$ image fields per staining). (D, E) frequencies of (D) T-cells and (E) neutrophils in the lungs after treatment with REM vs. vehicle ($n = 4$ mice per group). The box plots show minima, maxima, mean, 75, and 25 percentiles. (F) GESA transcriptomic analysis of lung RNA from vehicle group vs. REM-treated group. (Class A: vehicle group. Class B: REM-treated group). Data were presented as mean ± SEM. Data information: "$n$" indicates biological replicates. Data were presented as mean ± SD. Statistical significance was calculated by one-way ANOVA with Bonferroni correction (A) or two-tailed unpaired $t$-test (C–E). *$P \leq 0.05$, ***$P \leq 0.001$., ns non-significant.

