## [Peer Review File · EMBO Reports]

Novel immunomodulatory properties of adenosine analogs promote their antiviral activity against SARS-CoV-2

Giulia Monticone, Zhi Huang, Peter Hewins, Thomasina Cook, Oygul Mirzalieva, Brionna King, Kristina Larter, Taylor Miller-Ensminger, Maria Sanchez-Pino, Timothy Foster, Olga Nichols, Alistair Ramsay, Samarpan Majumder, Dorota Wyczechowska, Darlene Tauzier, Elizabeth Gravois, Judy Crabtree, Jone Garai, Li Li, Jovanny Zabaleta, Mallory Barbier, Luis Del Valle, Kellie Jurado, and Lucio Miele

Corresponding author: Giulia Monticone (gmonti@lsuhsc.edu)

Review Timeline:

Submission Date:	23rd Aug 23
Editorial Decision:	21st Sep 23
Revision Received:	12th Dec 23
Editorial Decision:	1st Feb 24
Revision Received:	30th Apr 24
Accepted:	14th Jun 24

Editor: Achim Breiling

Transaction Report:

Dear Dr. Monticone,

Thank you for the submission of your research manuscript to EMBO reports. I have now received the reports from the three referees that were asked to evaluate your study, which can be found at the end of this email.

As you will see, the referees think that the findings are of high interest. However, referees #2 and #3 have several comments, concerns, and suggestions, indicating that a major revision of the manuscript is necessary to allow publication of the study in EMBO reports. As the reports are below, and all the referee concerns need to be addressed, I will not detail them here.

Given the constructive referee comments, I would like to invite you to revise your manuscript with the understanding that all referee concerns must be addressed in the revised manuscript and in a detailed point-by-point response. Acceptance of your manuscript will depend on a positive outcome of a second round of review. It is EMBO reports policy to allow a single round of revision only and acceptance of the manuscript will therefore depend on the completeness of your responses included in the next, final version of the manuscript.

- 1) a .docx formatted version of the final manuscript text (including legends for main figures, EV figures and tables), but without the figures included. Figure legends should be compiled at the end of the manuscript text.
- 2) individual production quality figure files as .eps, .tif, .jpg (one file per figure), of main figures (up to 8) and EV figures. Please upload these as separate, individual files upon re-submission.

- 3) a complete author checklist, which you can download from our author guidelines (<https://www.embopress.org/page/journal/14693178/authorguide>). Please insert page numbers in the checklist to indicate where the requested information can be found in the manuscript. The completed author checklist will also be part of the RPF.

- 4) a complete author checklist, which you can download from our author guidelines

(<https://www.embopress.org/page/journal/14693178/authorguide>). Please insert page numbers in the checklist to indicate where the requested information can be found in the manuscript. The completed author checklist will also be part of the RPF.

5) that primary datasets produced in this study (e.g. RNA-seq, ChIP-seq, structural and array data) are deposited in an appropriate public database. If no primary datasets have been deposited, please also state this in a dedicated section (e.g. 'No primary datasets have been generated and deposited'), see below.

The accession numbers and database should be listed in a formal "Data Availability" section (placed after Materials & Methods) that follows the model below. This is now mandatory (like the COI statement). Please note that the Data Availability Section is restricted to new primary data that are part of this study. This section is mandatory. As indicated above, if no primary datasets have been deposited, please state this in this section

Data availability

8) Regarding data quantification and statistics, please make sure that the number "n" for how many independent experiments were performed, their nature (biological versus technical replicates), the bars and error bars (e.g. SEM, SD) and the test used to calculate p-values is indicated in the respective figure legends (also for potential EV figures and all those in the final Appendix). Please also check that all the p-values are explained in the legend, and that these fit to those shown in the figure. Please provide statistical testing where applicable. Please avoid the phrase 'independent experiment', but clearly state if these were biological or technical replicates. Please also indicate (e.g. with n.s.) if testing was performed, but the differences are not significant. In case n=2, please show the data as separate datapoints without error bars and statistics. See also: <http://www.embopress.org/page/journal/14693178/authorguide#statisticalanalysis>

9) Please add scale bars of similar style and thickness to all the microscopic images, using clearly visible black or white bars (depending on the background). Please place these in the lower right corner of the images themselves. Please do not write on or near the bars in the image but define the size in the respective figure legend.

10) Please also note our reference format:

12) We now use CRediT to specify the contributions of each author in the journal submission system. CRediT replaces the author contribution section. Please use the free text box to provide more detailed descriptions and do not provide your final manuscript text file with an author contributions section. See also our guide to authors:

<https://www.embopress.org/page/journal/14693178/authorguide#authorshippinguidelines>

13) Please remove the list of abbreviations from the manuscript. Please define each abbreviation the first time it is mentioned in the text.

14) We would encourage you to use 'Structured Methods', our new Materials and Methods format. According to this format, the Materials and Methods section should include a Reagents and Tools Table (listing key reagents, experimental models, software and relevant equipment and including their sources and relevant identifiers) followed by a Methods and Protocols section in which we encourage the authors to describe their methods using a step-by-step protocol format with bullet points, to facilitate the adoption of the methodologies across labs. More information on how to adhere to this format as well as downloadable templates (.doc or .xls) for the Reagents and Tools Table can be found in our author guidelines (section 'Structured Methods'):

15) Please order the manuscript sections like this, using these names:

Title page - Abstract - Keywords - Introduction - Results - Discussion - Materials and Methods - Data availability section - Acknowledgements - Disclosure and Competing Interests Statement - References - Figure legends - Expanded View Figure legends

I look forward to seeing a revised version of your manuscript when it is ready. Please let me know if you have questions or comments regarding the revision.

Please use this link to submit your revision: <https://embor.msubmit.net/cgi-bin/main.plex>

Yours sincerely,

Referee #1:

This manuscript presents evidence that the antiviral drug Remdesivir can serve as an adenosine A2A receptor antagonist in addition to its direct antiviral activities. The A2A receptor antagonism results in increased T cell numbers and virus clearance concomitant with a decrease in inflammatory neutrophils. The authors provide a convincing set of in silico, in vitro, and in vivo experiments to support the notion that Remdesivir and potentially other nucleoside analogue drugs not only directly target viral replication, but also enhance beneficial aspects of immunity while dampening inflammation. This is a novel and important contribution to the fields of antiviral drug and coronavirus research that could be leveraged in future drug design. The manuscript is well written and the conclusions are reasonable and not overstated. No specific concerns were noted.

Referee #2:

In this study Monticone et al study novel immunomodulatory properties of the nucleotide analogue remdesivir and related molecules. They make the intriguing hypothesis that remdesivir may have additional properties beyond its direct anti SARS-CoV-2 activity, stimulating a more effective immune responses against the virus. This is an intriguing study with provocative but plausible hypotheses that are supported by their data. I have some comments which I hope will improve the manuscript. I found it difficult to follow in places, particularly the experimental details, the statistical tests required are not always applied and the data are in some cases somewhat over-interpreted. However, it's a very interesting study and I hope the authors agree with my suggestions.

1. Could the authors reword the text for clarity considering the assay in Fig 2A.
2. "To further test this hypothesis, we used CD8+ T-cells proliferation, which we know is reduced in response to CGS as shown in Monticone et al., 202223, as a readout in a dose-response study with increasing doses of CGS against a fixed dose of GS (Fig. 2A)." is unclear and the axis label "response:suppression of T-cell proliferation (%)" is equally unclear. I don't think this is a ratio and I think they're simply measuring T-cell proliferation but please clarify. I don't understand why the text says "which we know is reduced in response to CGS" but the line goes up. The axis label should refer to the measurement, not the result. Is this experiment in vitro in cells or in mice?
3. CGS does not appear to lead to a reduction in CD56+ T-cells in Fig 6B as claimed in line 130. Needs stats to show this and I'm not sure the data support this statement.
4. Fig 2B, please describe the experiment with more details. Measured how? Treated cells or mice?

5. Line 131 "whereas treatment withZM or REM did not change these read readouts or rather reverse these effects." This statement doesn't make sense or describe the data effectively. Again some statistical testing is important here but how can ZM or Rem reverse the effect of CGS if they're tested in separate experiments, which I think they are based on the labelling. If they are added separately, then does ZM increase either NLR or CD69 T cell count. I think nothing happens, except for the Rem 10uM on cd69 count, which is very noisy. I'm looking at the data and reading the text and they don't match making this very confusing. Please clarify, with statistical support, what is happening.
6. The statistical tests applied to Fig 2-F should go further. Eg firstly assess whether CGS has had an effect (vehicle vs CGS, test 1) and then test whether Rem has reduced this effect (CGS vs CGS + REM or vehicle compared to CGS+REM now n.s). Mark non significant where appropriate, ie the effect of CGS has gone in the presence of REM. Eg in 2D I think vehicle compared to CGS+REM will be non-significant showing the effect of CGS in this assay is completely inhibited. I do not see how adding ZM and seeing nothing change one can say its an antagonist and then assume GS will do the same.
7. The authors then say that they know that T cell proliferation is reduced in the presence of CGS according to another study. But their own data here don't show that (Fig S6B) and they don't say whether they mean reduced after a stimulation or what that stimulation is. Its really hard to follow. Line 136, don't just tell us the result, please explain the experiment.
8. Line 143. I found it very confusing to read "we then asked if GS could....so we used REM...." GS is not REM, this is explained but not until line 148. Please reconsider for clarity.
9. In Fig 2C-G, shouldn't REM also be tested alone. If REM and CGS are having opposing effects independently of each other they may not be connected. This would be revealed by testing whether REM has the opposite effect of CGS even in the absence of CGS. If REM is acting by inhibiting the effect of CGS then REM will only have an effect in the presence of CGS. Right?
10. I must say the gel in 2G does not reflect the statements made or the measurements shown. I wonder if the data from the 3 blots mentioned in the legend should be plotted and errors shown to persuade us of this result. The gel shown seems to show loss of the band on CGS treatment and it does not come back with REM addition.
11. Is it possible to provide some quantitation for histopathology 3D-F, 5D-F?
12. How sure can we be that ZM has no direct antiviral effect? I think much of the data could be explained by ZM having direct antiviral activity, which is not impossible given its similarity to adenosine. Perhaps it inhibits polymerase as REM does.
13. In Fig 4A, B, what does control signify? These experiments are not explained well. CGS is described at suppressing proliferation but actually shows almost as much proliferation as control. Is the proliferation stimulated in the presence of CGS and the idea is that proliferation is slightly lower in the presence of CGS. Then surely CGS + ZM should be compared to control and found not to be significantly different. I find this experiment hard to follow because the plots are poorly labelled and poorly described and the experiment is not well described. I'm not sure why some measurements have an unstimulated bar and some do not. I'm not sure whether the ZM stimulates the cells alone or whether it has no effect on the stimulation represented by control. Please label which bars are stimulated with what for clarity. I appreciate that this might be a standard approach, and may be obvious for initiates, but if the authors would like a broad audience to follow this work, more detail is required.
14. I don't really understand the rationale for putting particular data in the supplementary. I expect all data important to support the conclusions to be in the main text and additional controls or analysis to be put in the supp. Please move the data necessary to evaluate the conclusions in the main text. Otherwise one is switching constantly between the supp and the main figs to follow the story.
15. Line 191, I don't think the data support the statement that ZM is effective in clearing CoV-2 in mice. The mice clear it anyway, ZM just makes this happen a little bit more efficiently. Please reconsider wording to more accurately represent the data.
16. Line 203, interpreting the experiments in infected mice with REM is tricky because of course the REM has antiviral effects. I think many immunologists reason that immune driven disease is due to viral deregulation of innate immunity and cytokine driven immune dysfunction. Thus reducing virus replication might trivially explain enhanced immunity. I see the authors point, but please conclude more conservatively. For example, explain the plausible effect of viral suppression on immune responses and use phrasing such as "line 203 These findings are consistent with the treatment promoting T-cell responses..." I agree that they are. But indicate may be a bit strong, given alternate explanations. Toning conclusions down is also recommended in the final part of this paragraph.
17. Fig 5 please quantitate imaging.
18. Do the transcriptomics really confirm the that REM acts on A2AR? They're consistent with, but confirm may be a bit strong.

You'd really need to show this using the molecule that does not have direct anti-viral activity or all these results might simply be the effect of inhibiting replication. It seems obvious to me that inhibiting replication is going to enhance immunity. If the authors want to strongly conclude on this they should perform these experiments with an inhibitor that has demonstrated no direct antiviral effect or show that another unrelated, but effective, antiviral does not do this. These controls are lacking. I think this is OK but the authors should not conclude so strongly without these controls. Line 235 is the language I'm looking for.

Referee #3:

Here the authors proposed that antiviral adenosine analogs, like remdesivir, have previously unrecognized immunomodulatory properties that contribute to their therapeutic activity. They show that GS-44152a, a metabolite of remdesivir acts as an adenosine A2A receptor antagonist. They propose a new and rationale strategy for the design of next-generation antiviral agents with dual - immunomodulatory and intrinsic - antiviral properties. This is a very interesting work but some experiments are required to show the direct involvement of A2A receptor subtype

The hypothesis is that GS compounds may interact with A2AR and exert immunomodulatory effects in addition to their intrinsic antiviral activity. To test this idea, they first used a computational molecular docking, to predict the interaction between adenosine analogs and A2AR. The result of the in silico analysis is that the more stable poses for the GS metabolite are similar to adenosine, making polar interactions with S277, H278, E169 and N253. Whilst E169 and N253 are common residues for agonists and antagonists, S277 and H278 are more of a signature of agonist binding, with only a few antagonists reported to interact with such residues.

In order to determine whether GS has A2AR-mediated effects, the authors treated mice with the A2AR agonist CGS-21680 (CGS), the A2AR antagonist ZM-241385 (ZM) or REM. As a result, the authors claim : « We observed an increase in the Neutrophils-To-Lymphocytes (NLR) ratio and reduction in the number of splenic CD69+ CD3+ activated T-cells in a dose-dependent manner in CGS treated mice (Supplementary Fig. 6A-B), in agreement with the suppressive effect of activating A2AR in T-cells; whereas treatment with the A2AR antagonist ZM or REM did not change these readouts or rather reversed these effects » The last part of the sentence is unclear . In vivo, such molecules may have many target (off-target), as does adenosine. One key experiment is missing here. The authors should run a simple cell based assay to define the pharmacological properties of the GS molecules. After performing expression in standard cell line, running Gs G protein assay and monitoring the increased of intracellular cAMP will allow to determine if GS is an antagonist at A2A receptor, as well as for other subtype. Such type of experiment is well documented in the literature. In addition, cAMP production of mouse CD8+ T cell after treatment of 1 μ M CGS or 1 μ M GS or combination of both are not very convincing, although I do appreciate the difficulty of performing such experiment.

Line 136 : « We found that GS reduces the efficacy (Emax) and increases the EC50 of CGS (Fig. 2A). These are characteristics of an irreversible antagonist, which is competitive (increases the EC50) and insurmountable (decreases the Emax). I suggest changing "increases the EC50" to "reduces/decrease" potency. In pharmacology, increased potency means better potency. The second sentence is more confusing to me. Very few ligands are irreversible if not any, except if they are covalently bound.

« In line with the results of our molecular docking, GS could potentially bind with high affinity to A2AR, thus preventing the binding of any agonist to the receptor ». This is very unlikely and can be tested by pre-incubating your samples with the GS molecule and then activating with agonist. A clear molecular pharmacology characterisation is required to validate the antagonist properties of GS. Also CGS is not an absolute selective molecule for A2A and the conclusion should be carefully made. So I am concerned about the physiological effect observed here, to be mediated by different adenosine receptor subtype and not only A2A. This story is very interesting but a clear demonstration that GS is an antagonist of A2A subtype is required.

As a minor comment, please label all residues on panels presenting AR models.

To the reviewers of EMBO Reports,

We thank you for your insightful comments and suggestions. Enclosed please find our replies to your comments and our revised manuscript. We have performed the experiments requested by the reviewers, including a new *in vivo* study with remdesivir-treated mice, GPCR protein assays in a standard A2AR-expressing cell line and primary T-cells, as well as plaque-forming assay to determine if the A2AR antagonist ZM has direct antiviral activity. We also improved the description of experiments, plot displays, revised and clarified the conclusions as recommended by reviewers. Specific replies to individual comments are listed below (in red):

Referee #2:

In this study Monticone et al study novel immunomodulatory properties of the nucleotide analogue remdesivir and related molecules. They make the intriguing hypothesis that remdesivir may have additional properties beyond its direct anti SARS-CoV-2 activity, stimulating a more effective immune responses against the virus. This is an intriguing study with provocative but plausible hypotheses that are supported by their data. I have some comments which I hope will improve the manuscript. I found it difficult to follow in places, particularly the experimental details, the statistical tests required are not always applied and the data are in some cases somewhat over-interpreted. However, it's a very interesting study and I hope the authors agree with my suggestions.

We thank the reviewer for his insightful comments, which have helped us to significantly improve the manuscript. We addressed the comments point-by-point in the following section.

1. Could the authors reword the text for clarity considering the assay in Fig 2A.

We rewrote the explanation of the experiment as follows (from line 139, Fig. 2B in the revised paper). "We first performed a dose-response study by treating primary mouse CD8+ T-cells, activated with anti-CD3/CD28, with increasing doses of CGS against a fixed dose of GS (Fig. 2B). In this assay, we used CD8+ T-cells proliferation as a readout (response), since we know that CGS reduces proliferation in activated CD8+ T-cells as shown in *Monticone at al., 2022*²³ and other works^{24,25,27}. As expected, we observed that CGS reduces T-cell proliferation in a dose-dependent manner (Fig. 2B). Importantly, we found that GS antagonizes CGS-mediated suppression of proliferation, reduces the efficacy (E_{max}) and the potency of CGS (Fig. 2B)."

2. "To further test this hypothesis, we used CD8+ T-cells proliferation, which we know is reduced in response to CGS as shown in Monticone at al., 202223, as a readout in a dose-response study with increasing doses of CGS against a fixed dose of GS (Fig. 2A)." is unclear and the axis label "response:suppression of T-cell proliferation (%)" is equally unclear. I don't think this is a ratio and I think they're simply measuring T-cell proliferation but please clarify. I don't understand why the text says "which we know is reduced in response to CGS" but the line goes up. The axis label should refer to the measurement, not the result.

Is this experiment in vitro in cells or in mice?

Thank you for pointing this out. We rewrote this section: please see point #1 for the revised explanation of the experiment or line 139, Fig. 2B in the revised paper. For clarity, we explained in the main text and the legend that the experiment was performed in vitro in primary mouse CD8+ T-cells. We modified the figure as requested: the figure now shows “Response: T-cell proliferation (%)” on the Y axis and the drug dose on the X axis; the trend line goes down (= T-cell proliferation is reduced) with increasing dosage of CGS and the trend is weaker when combined with GS.

3. CGS does not appear to lead to a reduction in CD56+ T-cells in Fig 6B as claimed in line 130. Needs stats to show this and I'm not sure the data support this statement.

We added the statistical significance to the figure (Fig. 2A in the revised paper), which shows no significant change with ZM and REM and a significant decrease in CD69+ T-cells with CGS (2 mg/kg).

4. Fig 2B, please describe the experiment with more details. Measured how? Treated cells or mice? We added more details in the explanation of these experiments (Line 145, Fig. 2C in the revised paper): we explained in the text, figure title and legend that the first experiment (left side plot, Fig.2C) was performed in an A2AR-expressing cell line treated with CGS, GS and ZM, and cAMP was measured using a luciferase assay; the second experiment (right side plot, Fig. 2C) was performed in CD8+ mouse T-cells treated with CGS and GS, and cAMP was measured using ELISA: “To further test whether GS binds to A2AR and has antagonistic properties, we measured the production of cyclic AMP (cAMP), the direct secondary mediator of A2AR³⁶, in an A2AR-expressing cell line in response to GS, ZM and CGS, using a luciferase assay (Fig. 2C). We found that GS antagonizes and reduces cAMP production induced by CGS, similarly to the A2AR antagonist ZM (Fig. 2C). Along the same line, we also showed that GS antagonizes the overproduction of cyclic AMP (cAMP) in response to CGS, reducing it to control (vehicle) levels, as measured by ELISA in primary mouse CD8+ T-cells, (Fig. 2C).”

5. Line 131 "whereas treatment withZM or REM did not change these read readouts or rather reverse these effects."

This statement doesn't make sense or describe the data effectively. Again some statistical testing is important here but how can ZM or Rem reverse the effect of CGS if they're tested in separate experiments, which I think they are based on the labelling. If they are added separately, then does ZM increase either NLR or CD69 T cell count. I think nothing happens, except for the Rem 10uM on cd69 count, which is very noisy.

I'm looking at the data and reading the text and they don't match making this very confusing. Please clarify, with statistical support, what is happening.

We thank the reviewer for this comment. We have clarified the description of the experiment and added statistical testing results. We agree that the figure was meant to show that there is no significant change in NLR and CD69+ T-cells with ZM and REM, and a significant change with CGS. Therefore, we added the statistics in the figure (Fig. 2A in the revised paper).

We also agree that the sentence in line 131 was confusing and we revised it to clarify our description of the results (from line 130 in the revised manuscript): “We observed a dose-dependent increase in the Neutrophils-To-Lymphocytes (NLR) ratio and reduction in the number of splenic CD69+ CD3+ activated T-cells in CGS-treated mice (Fig. 2A), in agreement with the suppressive effect of activating A2AR in T-cells; whereas treatment with the A2AR antagonist

ZM or REM did not change these readouts (Fig. 2A). A2AR antagonists, which block the A2AR, do not usually show significant effects in the absence of an A2AR agonist, similarly to what we observed for the known A2AR antagonist, ZM, and REM. Therefore, based on these observations, we hypothesized that REM metabolite, GS, could act as an A2AR antagonist, like ZM. To address this hypothesis, we tested whether GS can antagonize the A2AR agonist CGS in a number of *in vitro* and *in vivo* studies.”

6. The statistical tests applied to Fig 2-F should go further. Eg firstly assess whether CGS has had an effect (vehicle vs CGS, test 1) and then test whether Rem has reduced this effect (CGS vs CGS + REM or vehicle compared to CGS+REM now n.s). Mark non significant where appropriate, ie the effect of CGS has gone in the presence of REM. Eg in 2D I think vehicle compared to CGS+REM will be non-significant showing the effect of CGS in this assay is completely inhibited. I do not see how adding ZM and seeing nothing change one can say its an antagonist and then assume GS will do the same.

We added the requested statistics in the figure (Fig. 2E-G in the revised paper), we compared: vehicle vs. REM to show that REM does not have a significant effect in the absence of the agonist CGS; CGS vs vehicle to show that CGS has a significant effect; CGS vs. CGS+REM to show that REM reduces CGS-mediated effects.

7. The authors then say that they know that T cell proliferation is reduced in the presence of CGS according to another study. But their own data here don't show that (Fig S6B) and they don't say whether they mean reduced after a stimulation or what that stimulation is. Its really hard to follow. Line 136, don't just tell us the result, please explain the experiment.

We agree that this paragraph was unclear and we rewrote the explanation of the experiment and plot display (from line 139, Fig. 2B in the revised paper): please also see our responses to the reviewer's comment #1 and #2. We specified that the experiment was performed in T-cells activated with anti-CD3/CD28 (stimulation) and that CGS reduces T-cell proliferation in a dose-dependent manner in this experiment, in line with our previous work (from line 139, Fig. 2B in the revised paper): “We first performed a dose-response study by treating primary mouse CD8+ T-cells, activated with anti-CD3/CD28, with increasing doses of CGS against a fixed dose of GS (Fig. 2B). In this assay, we used CD8+ T-cells proliferation as a readout (response), since we know that CGS reduces proliferation in activated CD8+ T-cells as shown in *Monticone at al., 2022*²³ and other works^{24,25,27}. As expected, we observed that CGS reduces T-cell proliferation in a dose-dependent manner (Fig. 2B). Importantly, we found that GS antagonizes CGS-mediated suppression of proliferation, reduces the efficacy (E_{max}) and the potency of CGS (Fig. 2B).”

8. Line 143. I found it very confusing to read "we then asked if GS could....so we used REM...." GS is not REM, this is explained but not until line 148. Please reconsider for clarity.

We moved the explanation about why we used REM in the section preceding this experiment (line 129 in the revised manuscript)

9. In Fig 2C-G, shouldn't REM also be tested alone. If REM and CGS are having opposing effects independently of each other they may not be connected. This would be revealed by testing whether REM has the opposite effect of CGS even in the absence of CGS. If REM is

acting by inhibiting the effect of CGS then REM will only have an effect in the presence of CGS. Right?

We thank the reviewer for this insightful comment: we agree that it is important to show that REM does not have an effect on its own in the absence of the agonist CGS. Therefore, we added a group treated with REM alone in the *in vivo* experiment in Fig. 2 (Fig. 2D-G in the revised paper). In line with our hypothesis, the results show that REM has immunomodulatory effects only when against CGS and we mentioned this in line 168 in the revised manuscript: “In all the experiments, REM or GS did not have effects as single agents, but only when used to antagonize CGS.”

10. I must say the gel in 2G does not reflect the statements made or the measurements shown. I wonder if the data from the 3 blots mentioned in the legend should be plotted and errors shown to persuade us of this result. The gel shown seems to show loss of the band on CGS treatment and it does not come back with REM addition.

We added a plot with the quantification of 3 separate Western blots, showing reduction of NICD by CGS and rescuing of NICD by GS (Fig. 2H in the revised paper)

11. Is it possible to provide some quantitation for histopathology 3D-F, 5D-F? We added the quantifications. These are displayed in Fig.3 and Supplementary Fig. 8 (for Fig.5).

12. How sure can we be that ZM has no direct antiviral effect? I think much of the data could be explained by ZM having direct antiviral activity, which is not impossible given its similarity to adenosine. Perhaps it inhibits polymerase as REM does.

We thank the reviewer for this comment. We addressed this point in two ways, theoretically and experimentally: (1) The structure of ZM suggests that this compound lacks direct antiviral activity: compared to adenosine or GS, ZM conformation lacks a functional ribose moiety that allows incorporation in the viral genome. Please see the structure of ZM vs. GS in the figure below. (2) We performed a plaque-formation assay (a standard assay for testing antiviral drugs) to determine whether ZM has any direct antiviral activity against SARS-CoV-2 (Fig. 3A in the revised paper). In this assay, Vero monkey cells are infected with SARS-CoV-2, treated with the compound of interest and the viral burden (number of infecting viral particles) is determined after the treatment by counting the number of cells lysed by the virus (plaque—forming units, PFU). We found that ZM does not reduce the viral burden, suggesting that it has no direct antiviral activity. We described this experiment in line 177 in the revised paper: “We first needed to exclude that any antiviral effect observed in response to an A2AR antagonist may reflect direct antiviral activity. Therefore, we tested whether the A2AR antagonist, ZM, has any direct antiviral activity against SARS-CoV-2: we performed a plaque forming assay in Vero cells infected with SARS-CoV-2 and treated with different doses of ZM. We did not observe a reduction in the viral burden in Vero cells in response to ZM, thus suggesting that ZM does not have intrinsic antiviral effect against SARS-CoV-2 (Fig. 3A).”

13. In Fig 4A, B, what does control signify? These experiments are not explained well. CGS is described at suppressing proliferation but actually shows almost as much proliferation as control. Is the proliferation stimulated in the presence of CGS and the idea is that proliferation is slightly lower in the presence of CGS. Then surely CGS + ZM should be compared to control and found not to be significantly different. I find this experiment hard to follow because the plots are poorly labelled and poorly described and the experiment is not well described. I'm not sure why some measurements have an unstimulated bar and some do not. I'm not sure whether the ZM stimulates the cells alone or whether it has no effect on the stimulation represented by control. Please label which bars are stimulated with what for clarity. I appreciate that this might be a standard approach, and may be obvious for initiates, but if the authors would like a broad audience to follow this work, more detail is required.

We agree that this set of experiment was not clearly presented, and we made the requested changes as follows:

(1) “control” stands for cells receiving vehicle (DMSO). For clarity we changed “control” with “vehicle”.

(2) We changed the layout of the figure by organizing the data according to the readout instead of cell type to make it clearer and added a more detailed explanation in the figure legend.

(3) What we wanted to show with this set of experiments is the antagonism between CGS and ZM in T-cell functional readouts, similarly to what we showed for GS vs. CGS in Fig 2. Therefore, we statistically compared CGS vs vehicle to show CGS-mediated effects and CGS vs CGS+ZM to show the rescuing/antagonistic capacity of ZM. ZM does not usually have an effect on its own or it has a mild one, with the exception of Fig 4G,H, where it significantly decreases IL-10 and 17. Therefore, to better show these results we also compared ZM vs vehicle and CGS+ZM vs vehicle in Fig. 4G,H.

(4) We rewrote part of the explanation of the experiments to make it clearer (line 193 and figure legend in the revised paper): “In agreement with our previous work²³, we found that activation of A2AR by the A2AR agonist, CGS, downregulates the active form of Notch1, NICD - a key

regulator of T cell functions, including cytokine production^{23,40,41} - and suppresses proliferation, whereas treatment with ZM rescues NICD and proliferation in primary mouse CD8+ T-cells and CD4+ T-cells (Fig. 4A-D). Consistently, we found that CGS and ZM antagonize each other's effects on cytokine production: CGS suppresses IFN- γ , whereas ZM rescues IFN- γ production from CGS in CD8+ T-cells and Th1 CD4+ T-cells (Fig. 4E,F); CGS does not change IL-10 in Th2 CD4+ T-cells, while ZM reduces IL-10 when used as a single treatment or against CGS (Fig. 4G); CGS increases IL-17 in Th17 CD4+ T-cells, whereas ZM restores it to control (vehicle) level, antagonizing CGS (Fig. 4G, H)."

(5) When we measure T-cell proliferation we usually include an "unstimulated" control that is not activated with anti-CD3/28 antibodies. This control may be included to show that activation with anti-CD3/28 successfully triggers proliferation. Cells treated with CGS still proliferate more than the unstimulated ones because they are activated with anti-CD3/28, but to a weaker extent compared to vehicle-receiving cells. We agree with the reviewer that the unstimulated bars could be confusing for the general readership of the journal and we opted to remove the unstimulated bars from the figure. In the revised figure, all the bars show cells that were activated with anti-CD3/28, and we specified this in the figure legend.

14. I don't really understand the rationale for putting particular data in the supplementary. I expect all data important to support the conclusions to be in the main text and additional controls or analysis to be put in the supp. Please move the data necessary to evaluate the conclusions in the main text. Otherwise one is switching constantly between the supp and the main figs to follow the story.

We agree with this suggestion and we moved the most relevant data in the main figures: e.g. we moved NLR and CD69 T-cells plots from Supplementary Fig. 6 to main Fig 2A.

15. Line 191, I don't think the data support the statement that ZM is effective in clearing CoV-2 in mice. The mice clear it anyway, ZM just makes this happen a little bit more efficiently. Please reconsider wording to more accurately represent the data.

We changed the wording of the sentence from "promotes faster viral clearance" to "increases the efficiency of viral clearance" (line 186 in the revised paper).

16. Line 203, interpreting the experiments in infected mice with REM is tricky because of course the REM has antiviral effects. I think many immunologists reason that immune driven disease is due to viral deregulation of innate immunity and cytokine driven immune dysfunction. Thus reducing virus replication might trivially explain enhanced immunity. I see the authors point, but please conclude more conservatively. For example, explain the plausible effect of viral suppression on immune responses and use phrasing such as "line 203 These findings are consistent with the treatment promoting T-cell responses..." I agree that they are. But indicate may be a bit strong, given alternate explanations. Toning conclusions down is also recommended in the final part of this paragraph.

We thank the reviewer for providing this insightful comment. We think that our data support the idea of remdesivir having immunomodulatory properties through stimulating the immune

response: this is based on our findings that ZM, a drug with no intrinsic antiviral activity, stimulates antiviral immune responses in infected mice and remdesivir elicits immunomodulation in non-infected mice; however, we agree that there may be different explanations for the immunological effects we observed with remdesivir in infected mice. Therefore, following the reviewer's suggestion we toned down the language in the entire result section titled "Remdesivir has immune-modulatory effects analogous to A2AR antagonist ZM in promoting antiviral immune responses in SARS-CoV-2-infected mice.": we replaced the strongest terms such as, "indicate", "strongly indicating", "is effective in", "to confirm/confirming" "provide compelling evidence", with more conservative language like "suggests/suggesting", "may", "promotes", "to explore", "which supports the idea". Finally, we toned down the last part of this paragraph (line 258 in the revised paper): "Collectively, our findings support the hypothesis that REM may have immunomodulatory effects which are achieved through A2AR antagonism, and may be independent of its intrinsic antiviral activity."

We also added the insightful discussion point raised by the reviewer in the discussion section of the paper (line 309 in the revised paper): "Our data support the idea that REM has immunomodulatory effects mainly through A2AR antagonism of its metabolite GS. However, it is possible that part of the immunological effects observed in infected mice treated with REM may be a result of decreasing viral load, thus decreasing the immune deregulation induced by the infection. Studies in mice treated with ZM, a drug with no intrinsic antiviral activity, compared with mice treated with REM could be performed to better define what immunological effects are due to REM intrinsic antiviral activity versus immunomodulatory properties."

17. Fig 5 please quantitate imaging. We added the quantifications for Fig. 5 in Supplementary Fig. 8.

18. Do the transcriptomics really confirm that REM acts on A2AR? They're consistent with, but confirm may be a bit strong. You'd really need to show this using the molecule that does not have direct anti-viral activity or all these results might simply be the effect of inhibiting replication. It seems obvious to me that inhibiting replication is going to enhance immunity. If the authors want to strongly conclude on this they should perform these experiments with an inhibitor that has demonstrated no direct antiviral effect or show that another unrelated, but effective, antiviral does not do this. These controls are lacking. I think this is OK but the authors should not conclude so strongly without these controls. Line 235 is the language I'm looking for.

We toned down the language of this section, as explained in point #16: for example, we wrote that these results "suggest/are consistent with" remdesivir acting on A2AR, instead of using "confirm". Having found that ZM does not have intrinsic antiviral activity against SARS-CoV-2 (Fig. 3A, point #12), also further supports that the immunological effects observed with remdesivir may be, at least in a significant part, due to its immunomodulatory properties. However, we agree that further studies comparing ZM and REM are needed for a deeper understanding of REM immunological properties, and we mentioned this in the discussion (please see our response to point #16). We thank the reviewer for providing this interesting discussion point.

Referee #3:

Here the authors proposed that antiviral adenosine analogs, like remdesivir, have previously unrecognized immunomodulatory properties that contribute to their therapeutic activity. They show that GS-44152a, a metabolite of remdesivir acts as an adenosine A2A receptor antagonist. They propose a new and rationale strategy for the design of next-generation antiviral agents with dual - immunomodulatory and intrinsic - antiviral properties. This is a very interesting work but some experiments are required to show the direct involvement of A2A receptor subtype

We thank the reviewer for his insightful comments and suggestions which we think helped us improving the paper. We addressed the comments point-by-point in the following section.

The hypothesis is that GS compounds may interact with A2AR and exert immunomodulatory effects in addition to their intrinsic antiviral activity. To test this idea, they first used a computational molecular docking, to predict the interaction between adenosine analogs and A2AR. The result of the in silico analysis is that the more stable poses for the GS metabolite are similar to adenosine, making polar interactions with S277, H278, E169 and N253. Whilst E169 and N253 are common residues for agonists and antagonists, S277 and H278 are more of a signature of agonist binding, with only a few antagonists reported to interact with such residues.

In order to determine whether GS has A2AR-mediated effects, the authors treated mice with the A2AR agonist CGS-21680 (CGS), the A2AR antagonist ZM-241385 (ZM) or REM. As a result, the authors claim : « We observed an increase in the Neutrophils-To-Lymphocytes (NLR) ratio and reduction in the number of splenic CD69+ CD3+ activated T-cells in a dose-dependent manner in CGS treated mice (Supplementary Fig. 6A-B), in agreement with the suppressive effect of activating A2AR in T-cells; whereas treatment with the A2AR antagonist ZM or REM did not change these readouts or rather reversed these effects » The last part of the sentence is unclear . In vivo, such molecules may have many target (off-target), as does adenosine. One key experiment is missing here. The authors should run a simple cell based assay to define the pharmacological properties of the GS molecules. After performing expression in standard cell line, running Gs G protein assay and monitoring the increased of intracellular cAMP will allow to determine if GS is an antagonist at A2A receptor, as well as for other subtype. Such type of experiment is well documented in the literature. In addition, cAMP production of mouse CD8+ T cell after treatment of 1µM CGS or 1µM GS or combination of both are not very convincing, although I do appreciate the difficulty of performing such experiment.

We agree that the description of Supplementary Fig 6A,B was not sufficiently clear. The figure was meant to show that there is no significant change in NLR and CD69+ T-cells with ZM and REM, and a significant change with CGS. Therefore, we added the statistics in the figure (Fig. 2A in the revised paper). We also revised the writing to make the results description clearer (line 130 in the revised paper): “We observed a dose-dependent increase in the Neutrophils-To-Lymphocytes (NLR) ratio and reduction in the number of splenic CD69+ CD3+ activated T-cells in CGS-treated mice (Fig. 2A), in agreement with the suppressive effect of activating A2AR in

T-cells; whereas treatment with the A2AR antagonist ZM or REM did not change these readouts (Fig. 2A). A2AR antagonists, which block the A2AR, do not usually show significant effects in the absence of an A2AR agonist, similarly to what we observed for the known A2AR antagonist, ZM, and REM. Therefore, based on these observations, we hypothesized that REM metabolite, GS, could act as an A2AR antagonist, like ZM.”

We thank the reviewer for suggesting that we perform the cAMP experiment in a standard A2AR-expressing cell line. As shown in Fig. 2C (and line 145) in the revised paper, we treated a stable A2AR-expressing cell line (Abeomics, 14-522ACL, Cho cells) with GS or ZM or CGS alone or in combination and detected the production of cAMP using a luciferase assay (G protein assay, Promega cAMP Glow assay): we found that GS antagonizes and reduces the cAMP production induced by CGS, similarly to the effect observed in response to the known A2AR antagonist ZM. This is consistent with our hypothesis that GS acts as an A2AR antagonist and we thank the reviewer for suggesting this key experiment.

Line 136 : « We found that GS reduces the efficacy (E_{max}) and increases the EC_{50} of CGS (Fig. 2A). These are characteristics of an irreversible antagonist, which is competitive (increases the EC_{50}) and insurmountable (decreases the E_{max}). I suggest changing "increases the EC_{50} " to "reduces/decrease" potency. In pharmacology, increased potency means better potency. The second sentence is more confusing to me. Very few ligands are irreversible if not any, except if they are covalently bound.

« In line with the results of our molecular docking, GS could potentially bind with high affinity to A2AR, thus preventing the binding of any agonist to the receptor ». These is very unlikely and can be tested by pre-incubating your samples with the GS molecule and then activating with agonist. A clear molecular pharmacology characterisation is required to validate the antagonist properties of GS. Also CGS is not an absolute selective molecule for A2A and the conclusion should be carefully made. So I am concern about the physiological effect observed here, to be mediated by different adenosine receptor subtype and not only A2A. This tory is very interesting but a clear demonstration that GS is an antagonist of A2A subtype is required.

We agree with the reviewer that we did not have sufficient evidence to claim that GS is irreversible. To address this point, we welcomed the reviewer’s suggestion to perform an experiment in which we pre-incubated cells with GS before activating A2AR with the agonist CGS. If GS is irreversible, we would expect to observe a complete inhibition of CGS-induced cAMP production. We did this experiment in the A2AR-expressing cell line and used cAMP as a readout: we found that pre-incubation with GS counteracted CGS-induced cAMP, but did not abolish CGS effect completely (please see figure below), suggesting that GS has antagonistic activity, but is most likely not irreversible. We obtained similar results with ZM, which is known to be a reversible A2AR antagonist. Therefore, we removed this claim from the paper and we rewrote the relevant paragraph accordingly (line 143 in the revised paper): “Importantly, we found that GS antagonizes CGS-mediated suppression of proliferation, reduces the efficacy (E_{max}) and the potency of CGS (Fig. 2B). ~~These are characteristics of an irreversible antagonist, which is competitive (increases the EC_{50}) and insurmountable (decreases the E_{max}).~~”

Figure legend: A2AR-expressing cells (Cho cell line stably expressing A2AR, Abeomics) were pre-treated with GS, ZM for 30 minutes and cells were then washed to remove unbound drug (yellow, orange, red and dark red bars). CGS was added for 30 minutes alone (blue bar) or after pre-treatment with GS or ZM (red and dark red bars). Vehicle control indicates cells receiving DMSO. The plot shows cAMP production which was measured using a luciferase assay (Promega, cAMP-glow assay). Student's t-test was used to calculate the statistical significance between CGS and combination treatment samples.

Our data, including the newly added GPCR assay experiment in A2AR-expressing cell line, supports the idea that GS functions as an A2AR antagonist. We used the same GPCR assay in primary mouse CD8+ T-cells to further test whether the effect of GS is mainly on A2AR and not other adenosine receptors: we induced cAMP production in primary mouse CD8+ T-cells using selective agonists for A2AR, A2BR, A1, A3 and tested whether GS could antagonize cAMP production (line 151 and Supplementary figure 6A in the revised manuscript): “Finally, we tested GS against agonists of the different adenosine receptors to determine its selectivity towards A2AR: we treated primary CD8+ T-cells with selective agonists for A2AR (CGS), A2BR (BAY 60-6583), A1 ((±)-5'-Chloro-5'-deoxy-ENBA), or A3 (CI-IB-MECA) alone or in combination with GS and measured cAMP production using a luciferase assay (Supplementary Fig. 6A). We found that GS significantly antagonizes cAMP overproduction mediated by the A2AR agonist, CGS, but not by the other agonists (Supplementary Fig. 6A), suggesting that GS at the concentrations tested is selective for A2AR compared to the other adenosine receptors.” We agree with the reviewer that small molecule agonists may not be absolutely specific to individual targets. However, numerous studies show that adenosine receptor agonists are highly selective towards their target. For example, CGS is a very well-known A2AR-specific agonist, cited virtually in all papers that look at A2AR including Ohta and Sitkovsky in Nature 2001 (PMID: 11780065), Kjaergaard and Sitkovsky, JI 2018 (PMID: 29802128) and Giuffrida *et al.*, in Nature Communications 2021 (PMID: 34050151). Genetic and *in vivo* methods could be used to explore the analogs properties in future studies and we mentioned this in the discussion (line 315 in the revised paper): “In this study we provided evidence that GS is more selective towards A2AR compared to other adenosine

receptors. However, the selectivity and pharmacology characterization of GS and other analogs should be further explored, using genetic methods and *in vivo* in mice.”

As a minor comment, please label all residues on panels presenting AR models.

We labelled all residues in all figures with AR models.

Dear Dr. Monticone,

Thank you for the submission of your revised manuscript to our editorial offices. I have already forwarded the report from the two referees that I asked to re-evaluate your study, you will find again below. I also have received your point-by-point-response (further revision plan). The referees have remaining concerns, and also feels that some of their points have not been adequately addressed, indicating that the paper needs further revision. After looking through your revision plan, I decided to invite a final revised manuscript that addresses the remaining referee points as indicated in your revision plan (see below). Please also provide a detailed final point-by-point-response to these.

I think, however, that the plaque forming assay should be repeated/performed as suggested by referee #2, including a positive control.

- Please provide a more detailed and comprehensive title, mentioning antiviral agents and SARS-CoV-2.
- Please list the authors on the title page as in the submission system (first name, last name).
- Please provide the abstract written in present tense throughout.
- Please order the manuscript sections like this, using these names:
Title page - Abstract - Keywords - Introduction - Results - Discussion - Materials and Methods - Data availability section - Acknowledgements - Disclosure and Competing Interests Statement - References - Figure legends - Expanded View Figure legends
- The Expanded View format, which will be displayed in the main HTML of the paper in a collapsible format, has replaced the Supplementary information. You can submit up to 5 images as Expanded View. Please follow the nomenclature Figure EV1, Figure EV2 etc. The figure legend for these should be included in the main manuscript document file in a section called Expanded View Figure Legends after the main Figure Legends section. Additional Supplementary material should be supplied as a single pdf file labeled Appendix. The Appendix should have page numbers and needs to include a table of content on the first page (with page numbers) and legends for all content. Please follow the nomenclature Appendix Figure Sx, Appendix Table Sx etc. throughout the text, and also label the figures and tables according to this nomenclature.

- Please change the references to our journal format:

- Please add scale bars of similar style and thickness to all the microscopic images, using clearly visible black or white bars (depending on the background). Please place these in the lower right corner of the images themselves. Please do not write on or near the bars in the image but define the size in the respective figure legend.

- Please make sure that the number "n" for how many independent experiments were performed, their nature (biological versus technical replicates), the bars and error bars (e.g. SEM, SD) and the test used to calculate p-values is indicated in the respective figure legends (for main, EV and Appendix figures) of the final revised manuscript. Please also check that all the p-values are explained in the legend, and that these fit to those shown in the figure. Please provide statistical testing where applicable. Please avoid the phrase 'independent experiment', but clearly state if these were biological or technical replicates. Please also indicate (e.g. with n.s.) if testing was performed, but the differences are not significant. In case n=2, please show the data as separate datapoints without error bars and statistics. See also:

<http://www.embopress.org/page/journal/14693178/authorguide#statisticalanalysis>

If n<5, please show single datapoints for diagrams. Could statistics also be added to the diagrams in Figs. S7 A/B and S8 D/E? Moreover:

- Please define the annotated p values ***/** in the legend of figure 6a; supplementary figure 8c; as appropriate.
- Please indicate the statistical test used for data analysis in the legends of figure 6a; supplementary figure 8c.
- Please note that the box plots need to be defined in terms of minima, maxima, center, bounds of box and whiskers, and percentile in the legends of figures 2c, e-f; 5j; supplementary figures 8d-e.
- Please note that information related to n is missing in the legends of figures 3a, c-d, f-g; 5b; supplementary figures 8d-e.
- Although 'n' is provided, please describe the nature of entity for 'n' in the legends of figures 2c, e-g; 4b-h; 5c; supplementary

figures 6a; 7a-b.

- Please define the error bar in the legend of figure 6a.

- Please format the figure legends for main, EV and Appendix figures) according to our journal style. See the respective section in our guide to authors (please find the link below). Please separate each panel description by a line break and make sure that the panels are listed in alphabetic order. Moreover, please add to each legend a 'Data Information' section explaining the statistics used or providing information regarding replicates and scales.

- Please make sure that all figure panels are called out separately and sequentially. Presently, there seems to be no callout for Supplementary Fig. 9. Please check.

- It seems that in figure 5 panels D,E,F and G the lower images show parts of the upper images in higher magnification. Thus, these are zoomed images and they therefore require zoom boxes (magnification boxes) identifying the area shown in the lower images in the upper images. Please do that. All these images need scale bars (see above).

In addition, I would need from you:

- a short, two-sentence summary of the manuscript (not more than 35 words).

- two to four short (!) bullet points highlighting the key findings of your study (two lines each).

- a schematic summary figure as separate file that provides a sketch of the major findings (not a data image) in jpeg or tiff format (with the exact width of 550 pixels and a height of not more than 400 pixels) that can be used as a visual synopsis on our website.

Best

,

Referee #2:

I'm afraid that the authors have not satisfactorily answered key questions in my review. Although new experimental data has been supplied it is not at all compelling. This can be fixed. Please see comments below. Otherwise the authors have answered all the questions I raised and have improved the manuscript accordingly

1. The whole story continues to hinge on whether ZM itself has antiviral activity. In the case that it does then this could explain the effect on the immunology during viral infection. Reduced replication leads to reduced immune response rather than ZM directly impacting immune response and outcome (weight loss). However, the new experiment demonstrating that ZM does not have any antiviral effect is not compelling. One doesn't normally add inhibitor to the plaque assay, one uses plaque assay to determine the amount of virus in time points from a time course of infection plus/minus inhibitor. In the case a plaque assay is used as presented in the new Fig 3A then the experiment needs a positive control (remdesivir) to show that inhibitor can reduce plaque count in this assay. Furthermore the dose of virus used in the plaque assay seems to be very high. By definition an MOI of 1 (1000 infectious units on 1000 cells) would infect 66% of the cells, way too many to allow plaque counting. So I don't quite understand how this was done. Finally, remdesivir typically requires pre-treatment for effective antiviral activity. The assay has been designed, perhaps not deliberately, to minimise the chance of ZM having any antiviral effect. This needs to be a lot more compelling.

I suggest a time course. Low dose infection followed by treatment with remdesivir, ZM or vehicle and then time points of supernatant collection at eg 2hours, 12 hours, 24 hours 48 hours. Supernatant titre can be measured by plaque or PCR for genome. I advise adding REM and ZM for at least 8hrs as a pre-treatment. I also advise showing photos of any plaques recorded in the supplementary. REM/ZM may reduce plaque size as well as plaque number.

If REM shows a decent antiviral effect and ZM does not in the same assay then that will cover it.

2. For clarity add CGS to x axis in Fig 2B.

3. The specificity presented in Fig S6 is not entirely compelling as presented because the degree of activation is normalised out

in the plot. We should be able to see activation by the agonist or the reason that there is no antagonism may be due to there being no activation to inhibit in the first place. I would also expect the agonist to be titrated to allow a fair comparison. Using a single dose of agonists to compare antagonism is risky. If the degrees of activation are different they may explain antagonist activity, ie the data may reflect the specificity of the agonist, not the antagonist.
Also label agonists on plots in Fig S6A.

4. The authors may also measure viral replication in the mouse samples. Does ZM treatment reduce viral replication by manipulating the immune response. That is an important finding along with an effective demonstration that ZM itself does not have antiviral activity.

5. Typo line 164.

Referee #3:

The authors address my comments but I have to disagree with the authors on CGS21680 selectivity. First, CGS21680 is not a selective molecule for all adenosine receptor subtypes. I can agree maybe for A2B but not for A1 and A3 for which binding affinity show only 10 fold differences (Gao and Jacobson, 2006), which is rather poor in terms of selectivity. Secondly, "Giuffrida et al., in Nature Communications 2021 (PMID: 34050151)" did not use CGS21680 in their study as provided for a justification in the rebuttal.

Rather than claiming that CGS is selective in order to consolidate the conclusion that GS is an A2A selective antagonist, I would suggest the authors to tone down such statement and to indicate, as they propose, that this part would require more work to reach out such conclusion.

Importantly, material and method section is very weak on how the cAMP assays were performed to generate the data presented in supplementary figure 6. Sup F6 is difficult to read and not convincing in my opinion, since the standard deviations are large. In addition, there is no mock control and other high affinity antagonist as positive control. This requires clarification; it is not clear to me how all four receptors can give the same cAMP production upon agonist stimulation. Indeed, A1 and A3 couple to G protein Gi/o; agonist stimulation will impair cAMP production and the cAMP concentration should be lower than for A2A (A2A is Gs coupled).

Finally, as presented by the authors and discussed in this article, GS uses a very similar binding pocket as adenosine, engaging similar residues that are conserved across the receptor family. So adenosine being not selective, is it difficult to expect GS to be. Altogether, this needs clarification.

To the reviewers of EMBO Reports,

We thank you for your insightful comments and suggestions. Enclosed please find our point-by-point replies to your comments and our revised manuscript. We have performed one more experiment to determine if the A2AR antagonist ZM has direct antiviral activity and we toned down and clarified the conclusion about the selectivity of GS towards A2AR. Specific replies to individual comments are listed below (in red):

Referee #2:

I'm afraid that the authors have not satisfactorily answered key questions in my review. Although new experimental data has been supplied it is not at all compelling. This can be fixed. Please see comments below. Otherwise the authors have answered all the questions I raised and have improved the manuscript accordingly

1. The whole story continues to hinge on whether ZM itself has antiviral activity. In the case that it does then this could explain the effect on the immunology during viral infection. Reduced replication leads to reduced immune response rather than ZM directly impacting immune response and outcome (weight loss). However, the new experiment demonstrating that ZM does not have any antiviral effect is not compelling. One doesn't normally add inhibitor to the plaque assay, one uses plaque assay to determine the amount of virus in time points from a time course of infection plus/minus inhibitor. In the case a plaque assay is used as presented in the new Fig 3A then the experiment needs a positive control (remdesivir) to show that inhibitor can reduce plaque count in this assay. Furthermore the dose of virus used in the plaque assay seems to be very high. By definition an MOI of 1 (1000 infectious units on 1000 cells) would infect 66% of the cells, way too many to allow plaque counting. So I don't quite understand how this was done. Finally, remdesivir typically requires pre-treatment for effective antiviral activity. The assay has been designed, perhaps not deliberately, to minimise the chance of ZM having any antiviral effect. This needs to be a lot more compelling.

I suggest a time course. Low dose infection followed by treatment with remdesivir, ZM or vehicle and then time points of supernatant collection at eg 2hours, 12 hours, 24 hours 48 hours. Supernatant titre can be measured by plaque or PCR for genome. I advise adding REM and ZM for at least 8hrs as a pre-treatment. I also advise showing photos of any plaques recorded in the supplementary. REM/ZM may reduce plaque size as well as plaque number.

If REM shows a decent antiviral effect and ZM does not in the same assay then that will cover it.

We thank reviewer #2 for the opportunity to clarify our methodology in the plaque-forming assay and strengthen our manuscript with additional data. We addressed the reviewer's point by repeating the assay and titrating the supernatants by PCR (Fig. 3A), including a positive control (remdesivir), pre-treatment with the drugs and reducing the MOI to 0.05, as recommended by the reviewer. The assay shows reduction of titer by remdesivir, but not with ZM, supporting the hypothesis that ZM does not have intrinsic antiviral activity. We have rewritten the text in the Results and Methods sections to better reflect the experimental procedure. These are the edits in the manuscript:

Line 184 – results: *"We first needed to exclude that any antiviral effect observed in response to an A2AR antagonist may reflect direct antiviral activity. Therefore, we tested whether the A2AR antagonist, ZM, has any direct antiviral activity against SARS-CoV-2. We treated Vero cells*

infected with SARS-CoV-2 with ZM or REM. We did not observe a reduction in the viral burden in Vero cells in response to ZM, in contrast with REM, thus suggesting that ZM does not have intrinsic antiviral activity against SARS-CoV-2 (Fig. 3A).”

Line 370 – methods:

“In vitro infection and viral load via RT-qPCR.

Vero cells were maintained in Dulbecco’s modified Eagle’s medium (DMEM), supplemented with 10% FBS and 1% Pen Strep and plated in 6-well tissue culture plates at 1-million cells per well. Cells were allowed to adhere overnight in a CO₂ incubator at 37°C. The following day, the culture media was aspirated from each well and 2mL of fresh DMEM+drug were added. After an hour incubation, cells were infected with a multiplicity of infection (MOI) of 0.05 in DMEM+drug. Two hours later, virus was removed and washed with 1X Phosphate-Buffered Saline (PBS). Fresh DMEM+drug was added to mock or virus-exposed cells and incubated for 24 hours. 24 hours post viral exposure, the cellular layer within each well was collected in TRIzol and homogenized. Cellular homogenate was added to Trizol at 1:3 ratio and extracted using Zymo Direct-zol RNA kit following manufactures protocol. RNA was reverse transcribed and amplified using iScript cDNA Synthesis Kit (Biorad). Gene-specific primers for SARS-CoV-2 N gene (F:TTACAAACATTGGCCGCAA & R:GCGCGACATTCCGAAGAA) and human HPRT1 (F: CATTATGCTGAGGATTTGGAAAGG & R: CTTGAGCACACAGAGGGCTACA) with Power SYBR Green PCR Master Mix (Applied Biosystems) were used to amplify viral and cellular RNA by QuantStudio 3 Flex Real-Time PCR Systems (Applied Biosystems). The relative expression levels of target genes were calculated using HPRT1 RNA as an internal control.”

In the first revision, we also provided the structure of ZM in comparison with the one of GS to support the claim that ZM does not have intrinsic antiviral activity: We point out that ZM lacks a ribose moiety, which would allow conversion into a ribonucleotide and incorporation in the viral genome. Please see the structure of ZM vs. GS in the figure below.

2. For clarity add CGS to x axis in Fig 2B.

We added the label indicating CGS.

3. The specificity presented in Fig S6 is not entirely compelling as presented because the degree of activation is normalised out in the plot. We should be able to see activation by the agonist or the reason that there is no antagonism may be due to there being no activation to inhibit in the first place. I would also expect the agonist to be titrated to allow a fair comparison. Using a single dose of agonists to compare antagonism is risky. If the degrees of activation are different they may explain antagonist activity, ie the data may reflect the specificity of the agonist, not the antagonist.

Also label agonists on plots in Fig S6A.

We thank the reviewer for raising this point and we agree that this needs clarification. We worked on addressing this point in the first revision by providing experiments using agonists against different adenosine receptors in primary T-cells (Supp Fig. 6 in the previous revision). However, we feel that this point may not be addressed in a reasonable time and with current resources at the level suggested by the reviewers. We think it would be wise, as the reviewers also seem to propose, to remove the claim that GS is absolutely selective towards A2AR and the assays in Supplementary Fig. 6. Additionally, we further stressed that this question will be better addressed in future studies in the results and discussion sections, line 177 (results): "We have not explored the activity of GS on other adenosine receptors and this will be the focus of future studies.", line 331 (discussion): "In this study we provided evidence that GS acts on A2AR; however, we have not yet tested whether GS may have effects through other adenosine receptors. The selectivity and pharmacology characterization of GS and other analogs should be further explored, using genetic methods and *in vivo* in mice. These questions need to be addressed in the future". This does not hamper the conclusions of the manuscript, as we did show that GS acts, at least in part, through A2AR. If GS has effects on other adenosine receptors, in addition to A2AR, this will still support its immunomodulatory functions.

4. The authors may also measure viral replication in the mouse samples. Does ZM treatment reduce viral replication by manipulating the immune response. That is an important finding along with an effective demonstration that ZM itself does not have antiviral activity.

Thank you for this suggestion. We addressed the potential intrinsic antiviral activity of ZM in point #1.

5. Typo line 164.

This sentence was removed.

Referee #3:

The authors address my comments but I have to disagree with the authors on CGS21680 selectivity. First, CGS21680 is not a selective molecule for all adenosine receptor subtypes. I can agree maybe for A2B but not for A1 and A3 for which binding affinity show only 10 fold differences (Gao and Jacobson, 2006), which is rather poor in terms of selectivity. Secondly, "Giuffrida et al., in Nature Communications 2021 (PMID: 34050151)" did not use CGS21680 in their study as provided for a justification in the rebuttal.

Rather than claiming that CGS is selective in order to consolidate the conclusion that GS is an A2A selective antagonist, I would suggest the authors to tone down such statement and to indicate, as they propose, that this part would require more work to reach out such conclusion.

Importantly, material and method section is very weak on how the cAMP assays were performed to generate the data presented in supplementary figure 6. Sup F6 is difficult to read and not convincing in my opinion, since the standard deviations are large. In addition, there is no mock control and other high affinity antagonist as positive control. This requires clarification; it is not clear to me how all four receptors can give the same cAMP production upon agonist stimulation. Indeed, A1 and A3 couple to G protein Gi/o; agonist stimulation will impair cAMP production and the cAMP concentration should be lower than for A2A (A2A is Gs coupled).

Finally, as presented by the authors and discussed in this article, GS uses a very similar binding pocket as adenosine, engaging similar residues that are conserved across the receptor family. So adenosine being not selective, is it is difficult to expect GS to be. Altogether, this needs clarification.

We thank the reviewer for raising this point and we agree that this needs clarification. We worked on addressing this point in the first revision by providing experiments using agonists against different adenosine receptors in primary T-cells (Supp Fig. 6 in the previous revision). However, we feel that this point may not be addressed in a reasonable time and with current resources at the level suggested by the reviewers. We think it would be wise, as the reviewers also seem to propose, to remove the claim that GS is absolutely selective towards A2AR and the assays in Supplementary Fig.6. Additionally, we further stressed that this question will be better addressed in future studies in the results and discussion sections, line 177 (results): "We have not explored the activity of GS on other adenosine receptors and this will be the focus of future studies.", line 331 (discussion): "In this study we provided evidence that GS acts on A2AR; however, we have not yet tested whether GS may have effects through binding to other adenosine receptors. The selectivity and pharmacology characterization of GS and other analogs should be further explored, using genetic methods and *in vivo* in mice. These questions need to be addressed in the future.". This does not hamper the conclusions of the manuscript, as we did show that GS acts, at least in part, on A2AR. If GS has effects on other adenosine receptors, in addition to A2AR, this will still support its immunomodulatory functions.

Dr. Giulia Monticone
Louisiana State University Health Sciences Center
United States

Dear Dr. Monticone,

Thanks for the submission of your further revised manuscript to our editorial offices. I have now received the final report by referee #2, and as you will see, the referee now fully reports the publication of the study. Referee #3 did not respond to our invitations to re-review the study, but looking through your p-b-p-response, I consider his/her remaining concern as adequately addressed.

I am thus very pleased to accept your manuscript for publication in the next available issue of EMBO reports. Thank you for your contribution to our journal.

Yours sincerely,

Referee #2:

The authors have effectively resolved the points I raised. The story is now compelling and all questions I had resolved.
